# In vitro generation of functional murine heart organoids via FGF4 and extracellular matrix

Jiyoung Lee [1✉], Akito Sutani[1,2], Rin Kaneko[1], Jun Takeuchi[3], Tetsuo Sasano[4], Takashi Kohda[1,5], Kensuke Ihara[3], Kentaro Takahashi[3], Masahiro Yamazoe[3], Tomohiro Morio[2], Tetsushi Furukawa[3] & Fumitoshi Ishino [1✉]

Our understanding of the spatiotemporal regulation of cardiogenesis is hindered by the difficulties in modeling this complex organ currently by in vitro models. Here we develop a method to generate heart organoids from mouse embryonic stem cell-derived embryoid bodies. Consecutive morphological changes proceed in a self-organizing manner in the presence of the laminin-entactin (LN/ET) complex and fibroblast growth factor 4 (FGF4), and the resulting in vitro heart organoid possesses atrium- and ventricle-like parts containing cardiac muscle, conducting tissues, smooth muscle and endothelial cells that exhibited myocardial contraction and action potentials. The heart organoids exhibit ultrastructural, histochemical and gene expression characteristics of considerable similarity to those of developmental hearts in vivo. Our results demonstrate that this method not only provides a biomimetic model of the developing heart-like structure with simplified differentiation protocol, but also represents a promising research tool with a broad range of applications, including drug testing.

[1] Department of Epigenetics, Medical Research Institute, Tokyo Medical and Dental University (TMDU), Tokyo 113-8510, Japan. [2] Department of Pediatrics and Developmental Biology, Tokyo Medical and Dental University (TMDU), Tokyo 113-8510, Japan. [3] Department of Bio-Informational Pharmacology, Medical Research Institute, Tokyo Medical and Dental University (TMDU), Tokyo 113-8510, Japan. [4] Department of Cardiovascular Medicine, Tokyo Medical and Dental University (TMDU), Tokyo 113-8510, Japan. [5] Present address: Faculty of Life and Environmental Sciences, University of Yamanashi, Yamanashi 400-8510, Japan. ✉email: jlee.epgn@mri.tmd.ac.jp; fishino.epgn@mri.tmd.ac.jp

Cardiogenesis—the process by which the heart develops during embryonic development—involves differentiation of diverse cell types, including atrial and ventricular muscle cells known as cardiomyocytes (CMs), specific myocytes constituting conducting tissue such as Purkinje fibers and smooth muscle (SM) cells, and nonmuscle cells including endothelial cells (ECs) and neuronal cells. It also encompasses the self-organization of the morphological changes needed to form the mature heart, including the formation of the heart tube, looping of the heart tube and formation of the chambered heart with atria and ventricles[1–5]. As such, the cardiogenic process is under tight spatiotemporal control. Many researchers have sought to develop an in vitro system that mimics in vivo cardiogenesis in order to better understand the function of heart, and provide technical advantages for drug testing and model for cardiac pathology including precardiac organoids with first heart field (FHF) and second heart field (SHF) contributing to cardiac development[6]. Also, the potential for making hearts from pluripotent stem cells were suggested[7]. Although cardiac cell differentiation protocols have been extensively developed, including in vitro CM differentiation from pluripotent stem cells (induced pluripotent stem cells (iPSCs) and embryonic stem cells (ESCs))[7–11] and in vivo/in vitro direct reprogramming of CM-like cells from fibroblasts[12–14], these protocols typically produce clumps of cardiac cells that do not closely resemble those of an in vivo heart, at least at the higher structural level. Although matrix rigidity-modulation enabled improved the contractile activity of cultured EBs, the efficiency of contracting EBs (15%) has remained low[15]. Recently, bioengineered chamber-specific human cardiac tissues from CMs have been successfully formed. However, three-dimensional (3D) heart organoid generation itself has been hampered by the structural complexity of the heart.

For 3D culture methods using ESCs or adult stem cells, ECMs including Matrigel and laminin were used for in vitro organogenesis, including the development of the brain[16], stomach[17], kidney[18], and intestinal tissues[19]. FGF and FGFR1 are required for cardiac gene expression, particularly during SHF proliferation and chamber formation in cooperation with the Sonic hedgehog gene during heart development[20,21]. Therefore, it may be beneficial to consider the role of the extracellular matrix (ECM) environment and FGF signal, which may promote the self-organizing process reconstructing in vivo cardiogenesis. The mouse model system is effective because the embryonic period of mice is shorter than that of humans and advantageous because embryonic heart samples are easily obtained for comparison, and many ESC lines with mutations in genes related to heart development are available for comprehensive analysis.

Here, we report the generation of a heart organoid with atrium- and ventricle-like structures showing the ultrastructural, histochemical and gene expression profile similarities to embryonic hearts in the presence of the laminin-entactin (LN/ET) complex[22] in the ECM and exogenous fibroblast growth factor 4 (FGF4)[20] via consecutive morphological changes like self-organization and, therefore, without the requirement of complex differentiation protocols.

## Results

**FGF4 and LN/ET complex promote heart organoid generation.** In vivo cardiogenesis occurs via consecutive morphological changes that include cardiac crescent formation on embryonic day 7.5 (E7.5), the beating heart tube derived from ECs and CMs (E8), the looping heart tube on E9.5 (with a beating force or outflow tract (OFT)) and formation of a four-chambered heart (right atrium, RA; left atrium, LA; right ventricle, RV and left ventricle, LV)[1–5], exhibiting regionality starts around E10.5 and maturation becomes completed at E14.5 (Supplementary Fig. 1a).

We first thought to determine the minimal set of growth factors needed to promote cardiogenic differentiation in vitro and hypothesized FGF4 is important for a necessary condition for in vitro reconstruction of cardiogenesis, given that FGF and FGFR1 are required for heart development. Additionally, reanalysis of microarray data from Li et al.[23], (GEO accession No. GDS5003) indicated that *Lama1*, *Lamb1*, *Lamc1*, and *Nid1* (entactin) are predominantly expressed in the embryonic heart (Supplementary Fig. 1b). Thus, we hypothesized that the presence of the LN/ET complex, containing laminin α1β1γ1 (laminin-111)[24] from an Engelbreth−Holm−Swarm tumor, would provide an adequate ECM environment for heart formation.

To test whether the conditions that we selected were sufficient for in vitro heart development, we cultured intact embryoid bodies (EBs) derived from mouse ESCs (1000−5000 cells/EB) in the absence of leukemia inhibitory factor (LIF), in the presence of gelated LN/ET complex, which includes components of the ECM in connective tissue, and exogenous FGF4 for induction of initial CM proliferation and chamber formation by day 8. Also, given that BMPs play central roles in neural crest cells (NCCs) differentiation through the expression of neuronal-specific genes, such as *Phox2b*, *Ascl1/Mash1/Cash1*, *Phox2a*, *Gata2/3* and *Hand2*[25], cardiac autonomic nerves might be formed in the presence of BMP4. Therefore, additional supplementation of 6′ bromoindirubin 3′ oxime (BIO, a Wnt activator), bone morphogenetic protein 4 (BMP4), and LIF from day 9 to day 13. Under nonembedded ECM conditions with exogenous FGF4, EBs expectedly exhibited effective morphogenesis (Fig. 1a).

Beating heart organoids with multiple chambers (similar to embryonic hearts) were observed after the EBs had been cultured for more than 10 days (up to 15 days) under the above culture condition (Fig. 1b, Supplementary Movie 1). Motion vector prediction analysis of the generated heart organoids cultured for 6−7 days revealed typical contractile behavior with contraction and relaxation (Supplementary Fig. 1c) and an average beating rate of 95.4 beats/min (±12.6, $n = 4$). However, EBs cultured without the LN/ET complex failed to undergo morphological changes, such as the formation of heart tubes and chambered hearts (Supplementary Fig. 1d), suggesting that the LN/ET complex promotes a proper extracellular environment for heart development. EBs cultured with alternative ECM, Matrigel, also exhibited low efficiency of heart organoid generation with multiple chambers (8%, $n = 12$, Fig. 1c), suggesting that nonembedded LN/ET complex enables the decreased mechanical hindrance that would otherwise inhibit cellular movement and prevent adequate patterning and self-organization.

To test whether other FGF signals (i.e., FGF10 that facilitate the differentiation of CMs[26]) enable to generate heart organoid like FGF4, we applied FGF10 and FGF2 (bFGF) for heart organoid generation. FGF10 showed 42% (first experiment, 5/12) and 25% (second experiment, 3/12) of heart organoid generation efficiency (such as chamber formation, Fig. 1d), while FGF4 displayed 88% and 78% of heart organoid generation efficiency (Fig. 1d, right). In case of bFGF supplementation, heart organoid generation efficiency was 58% (7/12) comparable to 78% (7/9) of FGF4 (Supplementary Fig. 1e). These results may be explained by the fact that bFGF and FGF4 have high affinity for FGFR1(IIIc) but FGF10 does not have such affinity for FGFR1(IIIc)[20]. Therefore, we next determined whether FGF4-FGFR1 pathway is required for proper chamber formation in vitro culture system. Inactivation of FGFR1 with PD173074 (an FGFR1 inhibitor) prevented the cultured EBs from developing into heart organoids and resulted in excessive proliferation of monolayer cells, suggesting that the FGF4-FGFR1 pathway regulates the proliferative balance

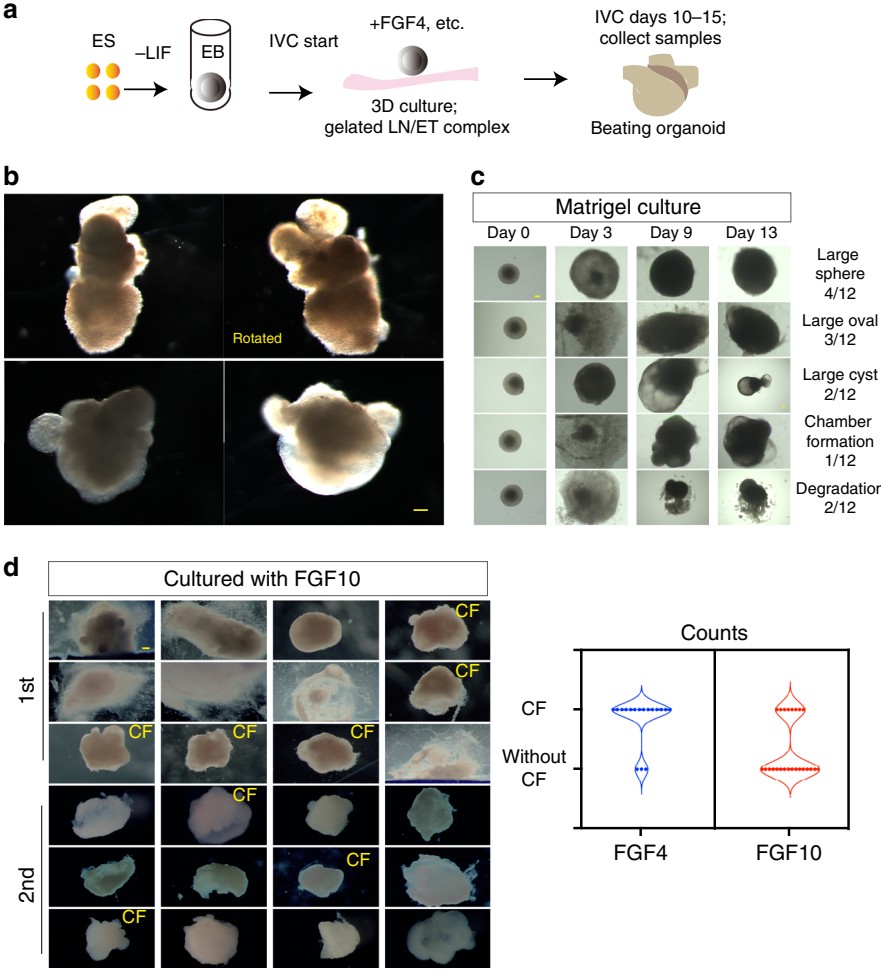

**Fig. 1 In vitro generation of developing heart-like structures. a** Experimental scheme to generate heart organoid (HO)s in vitro. EBs derived from ESCs were transferred to gelated LN/ET complex-coated chamber slides with HO medium containing FGF4. IVC in vitro culture. **b** Three representative beating HOs with multiple chambers generated using the above culture system after culturing for 10−11 days ($n =$ over 39, over six experiments). Scale bar: 100 µm. See also Supplementary Movie 1. **c** HOs tracking morphological changes over a time course during the culture with Matrigel. HOs were classified into four types according to their morphology ($n = 12$, two experiments). Chamber formation was hindered in the Matrigel culture. Scale bar: 100 µm. **d** Effect of FGF10 for HO formation. Twenty-four EBs were cultured with FGF10 instead of FGF4 on LN/ET complex (two experiments). Reduced efficiency (42%, $n = 12$ for the first experiment, and 25%, $n = 12$ for the second experiment) of chamber formation compared with the FGF4 condition (88%, $n = 8$ for the first experiment, and 78%, $n = 9$ for the second experiment) was shown after culturing for 15 days. Data are presented in a violin plot showing all points. Scale bar: 100 µm.

between CMs and other cells of the heart, including ECs, to ensure that the heart structure is organized correctly (Supplementary Fig. 1f, right).

The dramatic increase in the volume of the heart during development reflects the occurrence of myocyte mitosis in the embryonic heart. Notably, comparison between the cultured heart organoids ($n = 84$, all heart organoids; $n = 65$, looping heart tube or chamber formation + chamber formation) and embryonic hearts from E9.5 to E13.5 revealed that the heart organoids were larger than the embryonic hearts at E9.5 (Supplementary Fig. 2a). Although the sizes of generated heart organoids varied from 547 to 1957 µm, most heart organoids were over 700 µm (average = 835.1 µm, SEM = 29.9 µm, $n = 65$, Supplementary Fig. 2a, box plot). To determine whether the initial cell number was a limiting factor for effective heart organoid generation, we cultured EBs ($n = 11$) from a small number of ESCs (e.g., 500 cells). In these EBs, the size and beating ability of the resulting heart organoids significantly decreased, and the number of heart organoids present at the later morphogenetic steps (looping heart tube-like and chamber-like formation steps) was also reduced relative to

that of heart organoids from EBs derived from 1000 to 5000 cells. This result implies that the numbers of initial ESCs and differentiated beating CMs are critical factors; specifically, impaired beating resulted from a reduced number of CMs and led to defective morphogenesis (Supplementary Fig. 2b–d).

Next, we investigated whether in vitro heart organoids faithfully exhibit cardiac properties. The ultrastructural and structural compartments of heart organoids, as well as their development, were evaluated through comparison with embryonic mouse hearts. Transmission electron microscopy revealed that the heart organoids exhibited the CM-like ultrastructures that appear during embryogenesis, including sarcomere structures with Z-bands, mitochondria, and desmosomes (Fig. 2a). Additionally, further distinctive characteristics, such as the honeycomb structure of the sarcoplasmic reticulum (SR) (Fig. 2b, left) and intercalated disc (ID, and especially wavy ventricular myocyte IDs; Fig. 2b, right)[4], were observed in the heart organoids. The ID is a cardiac muscle-specific structure required for coordinated muscle contraction, indicating that these heart organoids might possess contractile cardiac muscle cell properties.

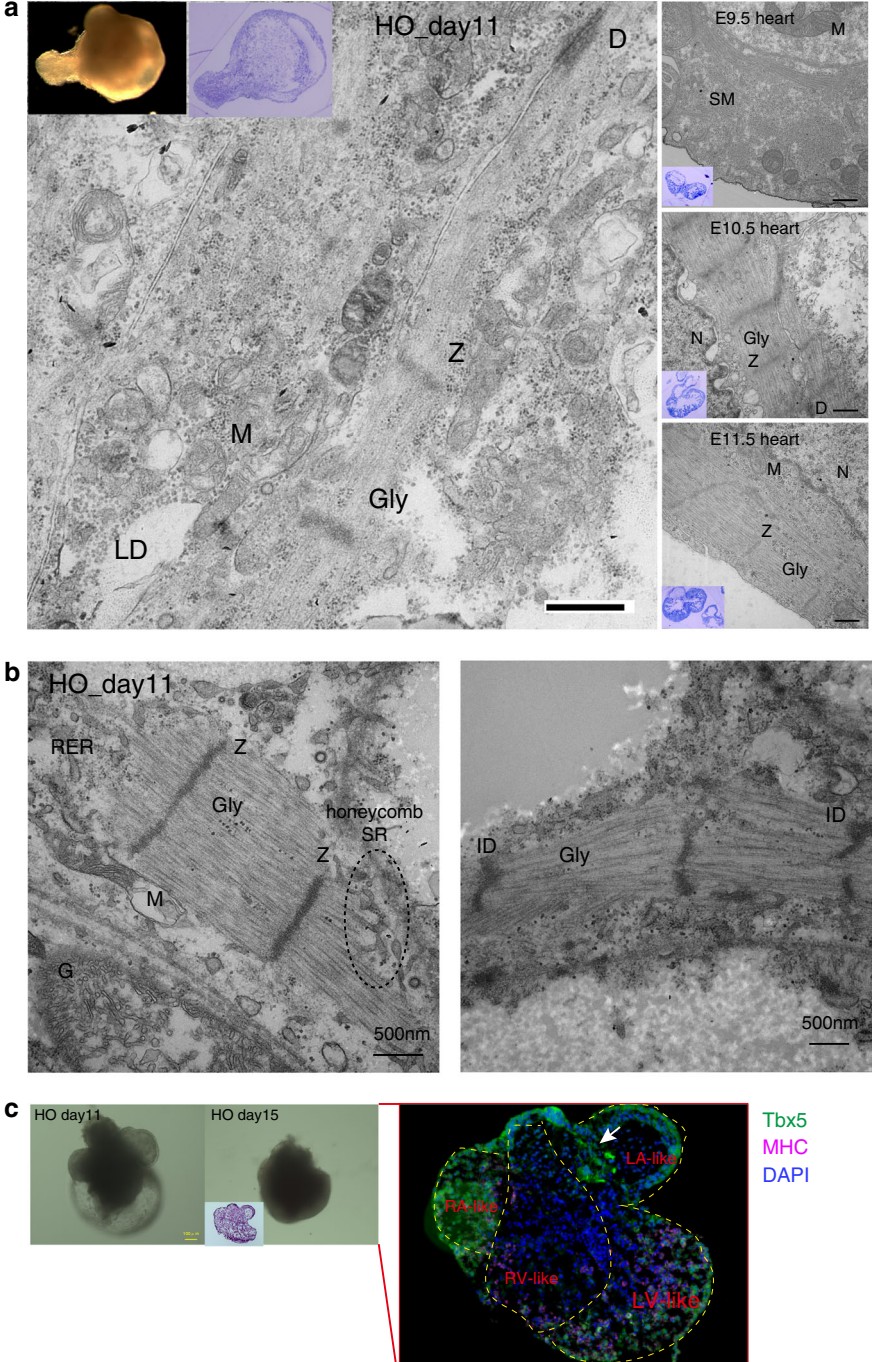

**Fig. 2 General features of heart organoid (HO)s. a** Ultrastructural analysis of the HO (insets, left: whole HO, right: a section stained with toluidine blue). Transmission electron microscopy of CM-like ultrastructures containing myofibrils with Z-bands (Z), mitochondria (M), glycogen granules (Gly) in myofibrils, lipid droplet (LD) and desmosomes (D). Scale bar: 400 nm. N nucleus, SM smooth muscle. Over $n = 5$, over three experiments. **b** Transmission electron microscopy of honeycomb sarcoplasmic reticulum (SR) structure (left) and wavy intercalated disc (ID; right) in ultrathin sections of CMs in the HOs. RER ribosomal endoplasmic reticulum, G Golgi complex. Scale bar: 500 nm; over $n = 4$, over three experiments. **c** Immunostaining to detect Tbx5 (green) and MHC (pink) (over $n = 7$, two experiments). The HO at day 11 and at day 15 (left panel, bright field, scale bar: 100 μm). Inset indicates hematoxylin and eosin-staining. Tbx5 expression (green) was positive in the left atrium (LA)-, right atrium (RA)- and left ventricle (LV)-like parts, MHC expression (pink) was strongly expressed in both LV- and RV-like parts, and a Tbx5-positive atria/ventricle (A/V) cushion-like structure was also detected (arrow). DAPI (blue) for nuclear staining.

To confirm whether the heart organoids (Fig. 2c, left) expressed cardiac transcription factors (TFs) and/or structural genes expressed by the in vivo embryonic heart, we assessed the expression of T-box transcriptional factor 5 (Tbx5, an important marker for proper development of both atria and the LV[27–31]) and SM-myosin heavy chain (SM-MHC, an SM marker). Notably, the spatial pattern of

Tbx5 expression in the heart organoids resembled that of in vivo embryonic hearts (E12.5H and E13.5H) after chamber formation (i.e., positive in the left atrium-, right atrium- and left ventricle-like parts) and also detected SM-MHC expression (Fig. 2c, right).

Taken together, these results suggested that FGF4 and the gelated LN/ET complex are required for effective in vitro

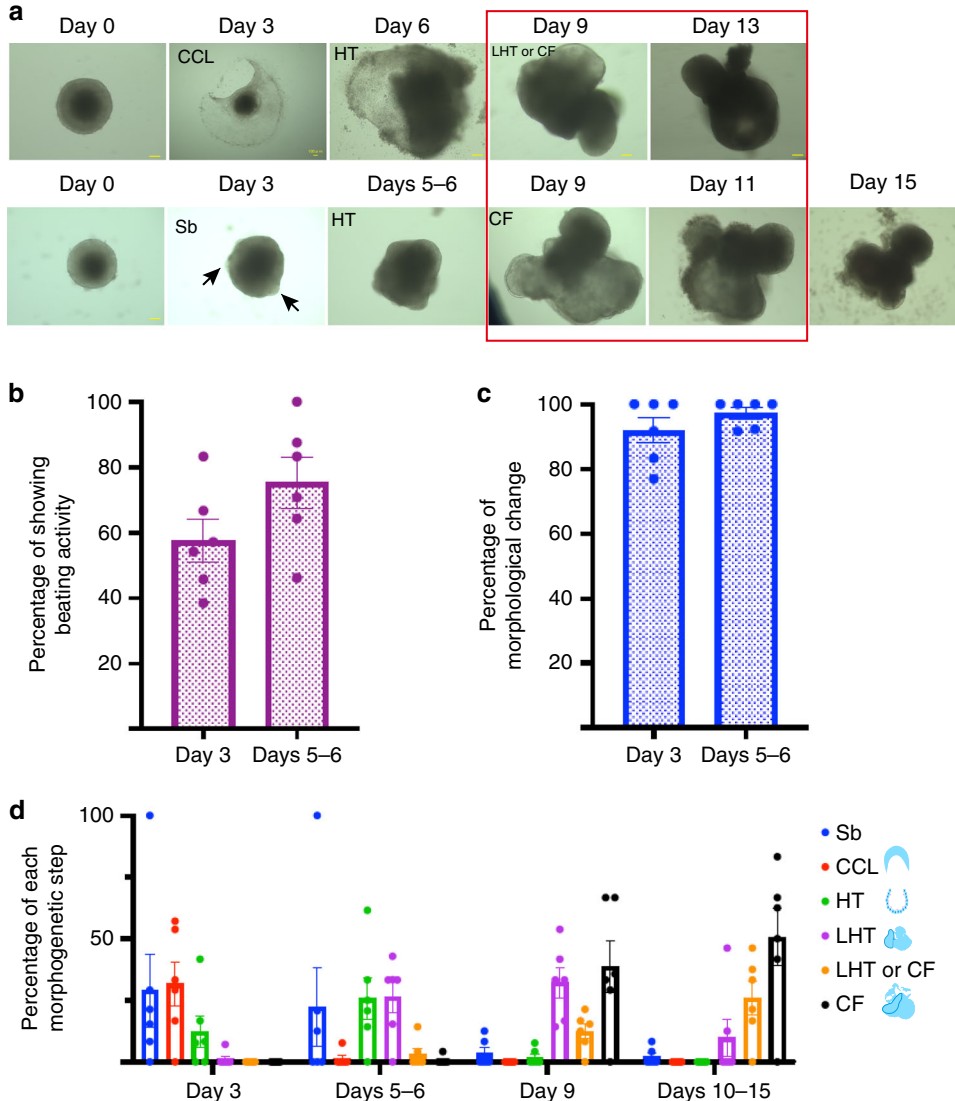

**Fig. 3 Different morphological stages during the in vitro cardiogenesis. a** Heart organoid (HO) tracking over a time course (over *n* = 84, over six independent experiments). The cardiac-crescent-like structures (CCL, top) or sphere with buds (Sb; similar to a small heart tube with polarity, bottom) appeared within 3 days of culture. Arrows indicate buds in a Sb. These structures underwent continuous morphological changes into heart tubes (HT), looping heart tubes (LHT) and chambered hearts (scale bar: 100 μm). The HOs were classified as being in the cardiac-crescent-like, Sb, heart tube, looping heart tube, or chamber-formation (CF) stage. **b** Half of the cultured HOs showed spontaneous beating movement within 3 days of culture (57.6%). The number of beating HOs increased at day 6 (75.4%; *n* = 84, six independent experiments). See also Supplementary Movie 4. **c** Morphological changes in the EBs appeared within 3 days of culture. At day 3, EBs with a spherical shape prior to culture had transformed into cardiac-crescent-like structures or Sbs (92.0%). Most EBs showed morphological changes until day 6 (97.3%). **d** Frequency of HOs with certain morphological characteristics according to the culture period. The majority of the HOs were classified as being in the cardiac-crescent-like stage (red bar) at day 3 and transformed to the heart tube and looping heart tube stages (light green and purple, respectively, days 5−6), looping heart tube and chamber-formation stages (purple and black, respectively, day 9) and finally to the chamber-formation stage (black) at days 10−15. Data are presented as mean values ± SEM in (**b–d**).

generation of heart organoids with similar cardiac properties to those of embryonic hearts.

**Morphological changes during in vitro cardiogenesis process.** Notably, the cell differentiation that occurred in the heart organoids reflected the morphogenetic changes during embryonic heart development in vivo. The formation of a cardiac-crescent-like structure, heart tube, and looping heart tube with chamber formation occurred in a defined order (Fig. 3a, top). The only exception was the appearance of a sphere with a bud (Sb), demonstrating a certain degree of polarity in in vitro cultured structure (Fig. 3a, bottom).

At E8.0 [1,3,5], cardiac precursor cells from the FHF migrate to the midline to form a linear beating heart tube containing CMs. This process is consistent with the spontaneous beating movement detected in the cultured heart organoids on day 3 (57.6%) and on day 6 (75.4%), indicating that the heart organoids also have functional capacity (Fig. 3b and Supplementary Movie 2) during the early developmental stages.

In terms of morphogenesis, most (92.0%) sphere-shaped EBs exhibited morphological changes on day 3, including the development of cardiac-crescent-like structures from outgrowths of EBs (Fig. 3a, c, and Supplementary Movie 3). By day 6, nearly all (97.3%) EBs had completed these changes (Fig. 3a, c), and by days 10−15, had shifted to the heart tube/looping heart tube,

looping heart tube/chamber and eventually to the chamber-formation stages, although approximately 20% of heart organoids exhibited Sb morphology that did not change by days 10−15. Most of the heart organoids that showed the heart tube morphology on days 5−6 proceeded to the looping heart tube structure (two ventricles) or chamber formation (ventricles and atria) by day 9. In the presence of FGF4, this developmental path of heart organoids resembled that of the embryonic heart in vivo (Fig. 3a, d).

Because FHF and SHF contribute to cardiac growth and morphogenesis[6], it should be determined whether the heart organoids mimic heart development even from an early gestation stage in terms of specification of FHF and SHF. Then, we investigated the histology of the cardiac-crescent-like structures ($n = 8$) and the heart organoids (over $n = 16$) to determine whether these heart organoids expressed two cardiac-specific TFs that are required for the embryonic heart formation and that are reliable gene markers of the cardiac crescent during early development (Tbx5, and the homeodomain-type transcription regulator Nkx2-5 [32]). At day 3 of EB culture with the gelated LN/ET complex, Tbx5 (FHF-like, Fig. 4a) and Nkx2-5 (SHF-like, Fig. 4a) expression was observed in the appropriate regions of the EBs, similar to the cardiac-crescent-like expression in the E7.5 heart in mice. In addition, the broad expression of Nkx2-5 and the restricted expression of Tbx5 in the cultured hearts resembled those observed in E11.5 hearts (Fig. 4b).

To ascertain that this system can be used to analyze in vitro cardiogenesis process, we performed the heart organoid generations using Tbx5 and Isl1 (Islet 1, a marker for proper looping during heart development[33]) knockout (KO) ESCs generated by the CRISPR/Cas9 system (Supplementary Fig. 3a). Expectedly, three Tbx5 KO ESC lines exhibited delayed morphogenesis including cardiac-crescent-like and heart tube formation or failure of heart organoid formation (no chamber formation for Tbx5 KO #4 and # 7, and only one chamber formation for Tbx5 KO #17), while chamber formation usually occurred at day 9 in case of WT ESCs (Supplementary Fig. 3b). Also, only one of three Isl1 KO ESC lines (#6) showed at most 12% ($n = 12$) of chamber formation at day 9 (Supplementary Fig. 3c), suggesting that this in vitro model also well reflects the known cardiac perturbation by deletion of Tbx5 and Isl1 during heart development in vivo. To further confirm whether cardiac TFs and other cardiac markers are normally expressed in the early morphogenetic stages of Tbx5 KO- and Isl1 KO-derived heart organoids, we examined Tbx5 and Nkx2-5 expressions at day 3 and Tbx5 and cTnT expressions at day 6. Expectedly, in the Tbx5 KO-derived heart organoids, Tbx5 expression was distinctively decreased and the typical location of Tbx5 and Nkx2-5 was disrupted although Nkx2-5 was expressed at day 3 (Fig. 4c, left). In the case of Isl1 KO-derived heart organoids, the expression of Tbx5 and Nkx2-5 was not clearly separated at day 3 (Fig. 4c, right). Moreover, both KO-derived heart organoids showed delayed morphogenesis such as cardiac-crescent-like structure and also exhibited weak expression of cTnT[34] and disorganization of cardiac fibers in cTnT-positive CMs compared to those of control heart organoids at day 6 (Fig. 4d). From these results, we conclude that the heart organoids can emulate cardiogenesis from the early gestation to mid-gestation stages, in terms of morphogenetic changes and expression of cardiac TF.

**Evaluation of the cell components of heart organoids**. In addition to cardiac muscle cells, cardiogenesis also depends on the proper organization of noncardiac muscle cells—including SM (in blood vessel walls) and ECs. Therefore, to determine whether the heart organoids contained SMs, we examined the

expression of α-smooth muscle actin (αSMA; SMC marker) and cTnT respectively. The expression of these markers was localized to discrete layers: strong expression of cTnT in the ventricle-like structures of the heart organoids, similar to the expression patterns detected in E10.5 heart and expression of αSMA inside the cTnT layer in the heart organoids (Fig. 5a). Interestingly, the Tbx5 KO-derived heart organoids showed decreased expression of αSMA and cTnT while the Isl1 KO-derived heart organoids exhibited disorganized expression pattern of αSMA, i.e., outside the cTnT layer (Supplementary Fig. 4a). In addition, we detected platelet/endothelial adhesion molecule 1 (PECAM; CD31, a representative EC marker[35] that is expressed in the inner membranes of primitive heart tubes and in vascular ECs in multichambered hearts) in the heart organoids (Fig. 5b), suggesting the presence of endocardial cell-layers in the heart organoids.

Further characterization of the heart organoids with series of cardiac markers GATA4 (required for heart tube formation)[36], cTnT, Connexin43 (Cx43, GJA1)[37], Cx40 (GJA5)[37] and Cx45 (GJC1)[37] indicated that the heart organoids have similar cardiac features to embryonic hearts (Fig. 5c–f). Cardiac connexins, Cx43 (in atrial and ventricular working myocardial cells), and Cx45 (localized in His-bundle and bundle branches) are gap junction proteins required for impulse propagation in the heart[37]. In addition, Cx40 is mainly expressed in atria of the heart. Notably, the differential localization of three connexins indicates the presence of different cell types with gap junctions and the expression pattern of Cx40 suggests the existence of atrial muscles with proper gap junction proteins in the heart organoids (Fig. 5d, e). In addition, the population of cTnI-positive and Nkx2-5-negative cells (white arrows, Supplementary Fig. 4b) suggests the presence of sinus node in the heart organoids like the embryonic hearts. No expression of a liver marker, α1 fetoprotein (AFP), also suggests the absence of functional endodermal parts in the heart organoids (Fig. 5f).

The transient receptor potential melastatin 4 (TRPM4) protein is a member of the TRP channel family and plays a role in mammalian cardiac electrical activity. TRPM4 is highly expressed in subendocardial bundles of heart Purkinje fibers[38–41], which transmit electrical impulses from the atrioventricular node bundle to CMs (working myocytes)[4]. To determine whether the heart organoids contained Purkinje fibers, we examined the expression of TRPM4 in both heart organoids and in vivo embryonic hearts. Immunofluorescence analysis revealed that TRPM4 was expressed in all heart organoids (100%, $n = 14$, Fig. 6a). In E9.5 hearts, there is little quantitative difference between TRPM4-positive Purkinje fiber-specific myocytes and cTnT-positive working myocytes, but working myocytes are clearly structured in E10.5 hearts by proliferation and differentiation for specification[40].

Purkinje-specific myocytes in Purkinje fibers differ ultrastructurally from working (ordinary) myocytes in the ventricular myocardium[42]. Accordingly, we identified specialized cells displaying ultrastructural features of Purkinje fibers—including wide Z bands, abundant intermediate filaments, the presence of M bands, and/or sparse myofibrils[4]—in the cultured heart organoids (Fig. 6b). These features indicate that these myocytes were Purkinje-specific cells, and not working (ordinary) myocytes, which have well-structured sarcomeres.

In mature mammalian hearts, a tetramer of type 2 ryanodine receptor (RYR2; ligand-activated $Ca^{2+}$ release channel) on the SR forms an SR/Transverse (T) tubule junction by binding with a voltage-gated $Ca^{2+}$ channel/dihydropyridine receptor (DHPR). RYR2 activation is required for excitation−contraction coupling in the cardiac muscle[43]. To determine whether heart organoids are associated with SR/T tubule junction, we examined the expression of RYR2 in heart organoids. We found that RYR2

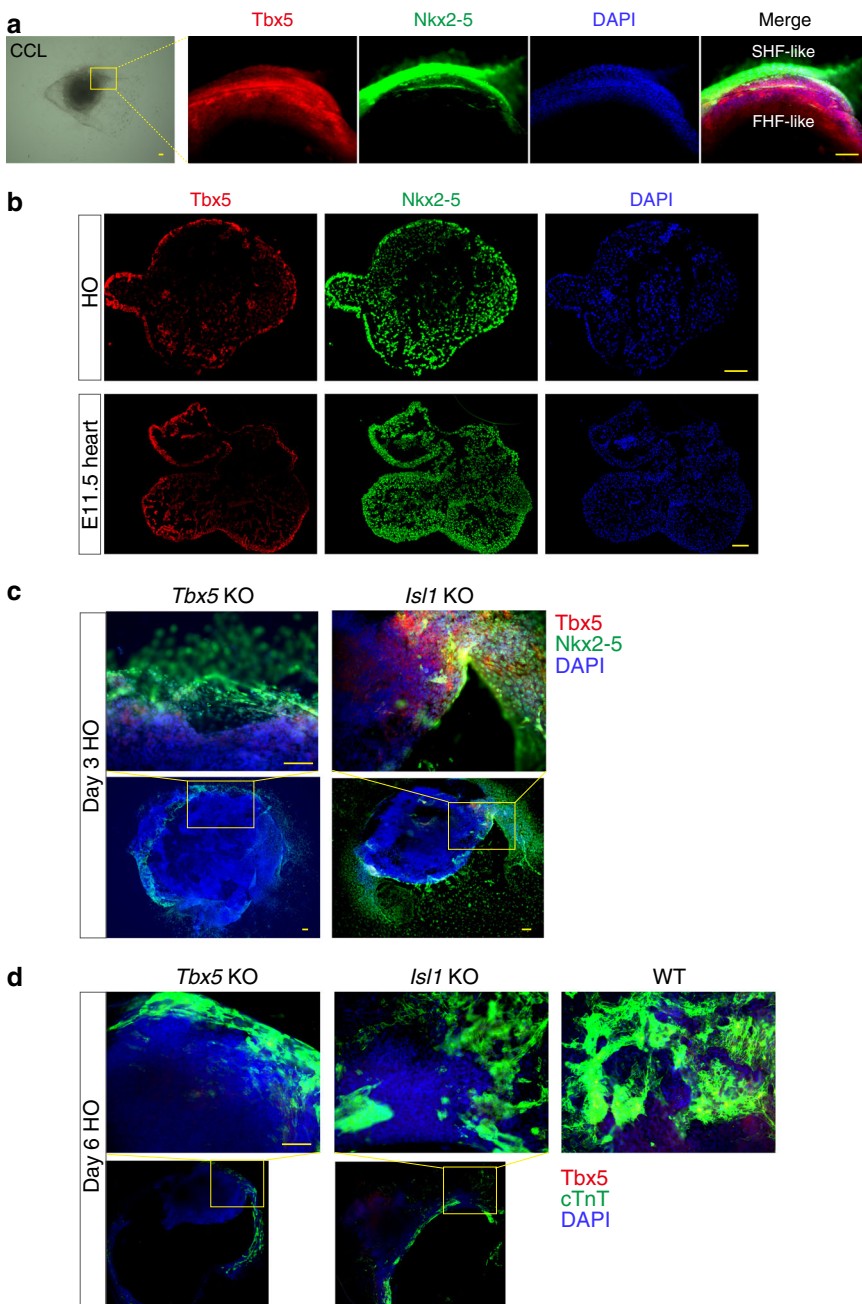

**Fig. 4 Expressions of cardiac transcription factors (TFs) and its perturbation in heart organoid (HO)s. a** Immunofluorescent staining to detect Nkx2-5 (green) and Tbx5 (red) in cardiac-crescent-like structures at day 3 of HO culture ($n = 8$, two experiments). Scale bar: 100 μm. **b** Immunofluorescent staining to detect Nkx2-5 (green) and Tbx5 (red) in HOs was cultured for more than 9 days. The representative HO ($n = 13$ out of $n = 16$) showed expression similar to that in embryonic hearts (E11.5). Scale bar: 100 μm. **c, d** Recapitulation of cardiac perturbation during in vitro HO formation from *Tbx5* KO- and *Isl1* KO-ESCs. **c** Immunofluorescent staining to detect Nkx2-5 (green) and Tbx5 (red) at day 3 of HO culture from derived *Tbx5* KO- ($n = 17$, left) and *Isl1* KO-ESCs ($n = 16$, right). In the *Tbx5* KO-derived HO, delayed or abnormal morphogenesis was observed with decreased Tbx5 expression and disrupted typical location of Tbx5 and Nkx2-5. In the case of *Isl1* KO-derived HOs, the expression of Tbx5 and Nkx2-5 was not clearly separated. Two experiments. Scale bar: 100 μm. **d** Immunofluorescent staining to detect cTnT (a CM marker, green)/αSMA (an SM marker, red) at day 6 of HO culture from derived *Tbx5* KO- ($n = 16$, two experiments), *Isl1* KO- ($n = 17$), and WT-ESCs ($n = 9$). The weak expressions of cTnT and/or αSMA compared to those of control (WT) HOs at day 6. Scale bar: 100 μm.

were expressed in the cultured heart organoids (100%, $n = 13$), while little expression of RYR2 was detected in E12.5H (Fig. 6c).

In the cardiogenesis, several types of cardiac cells and neuronal elements are differentiated from undifferentiated cells. To assess whether undifferentiated stem cells remained in the heart organoids cultured for 11 days, we performed immunostaining assays to detect the pluripotent stem cell marker Oct3/4. Concerning undifferentiated stem cell markers the heart organoids did not exhibit any Oct3/4 expression, in contrast to the EBs prior to culture, indicating that no pluripotent stem cells remained in the 11-day cultured heart organoid (Supplementary Fig. 5a), whereas Nestin which is expressed in the fetal heart as well as in brain tissue was detected in both of the heart organoids and E12.5 hearts.

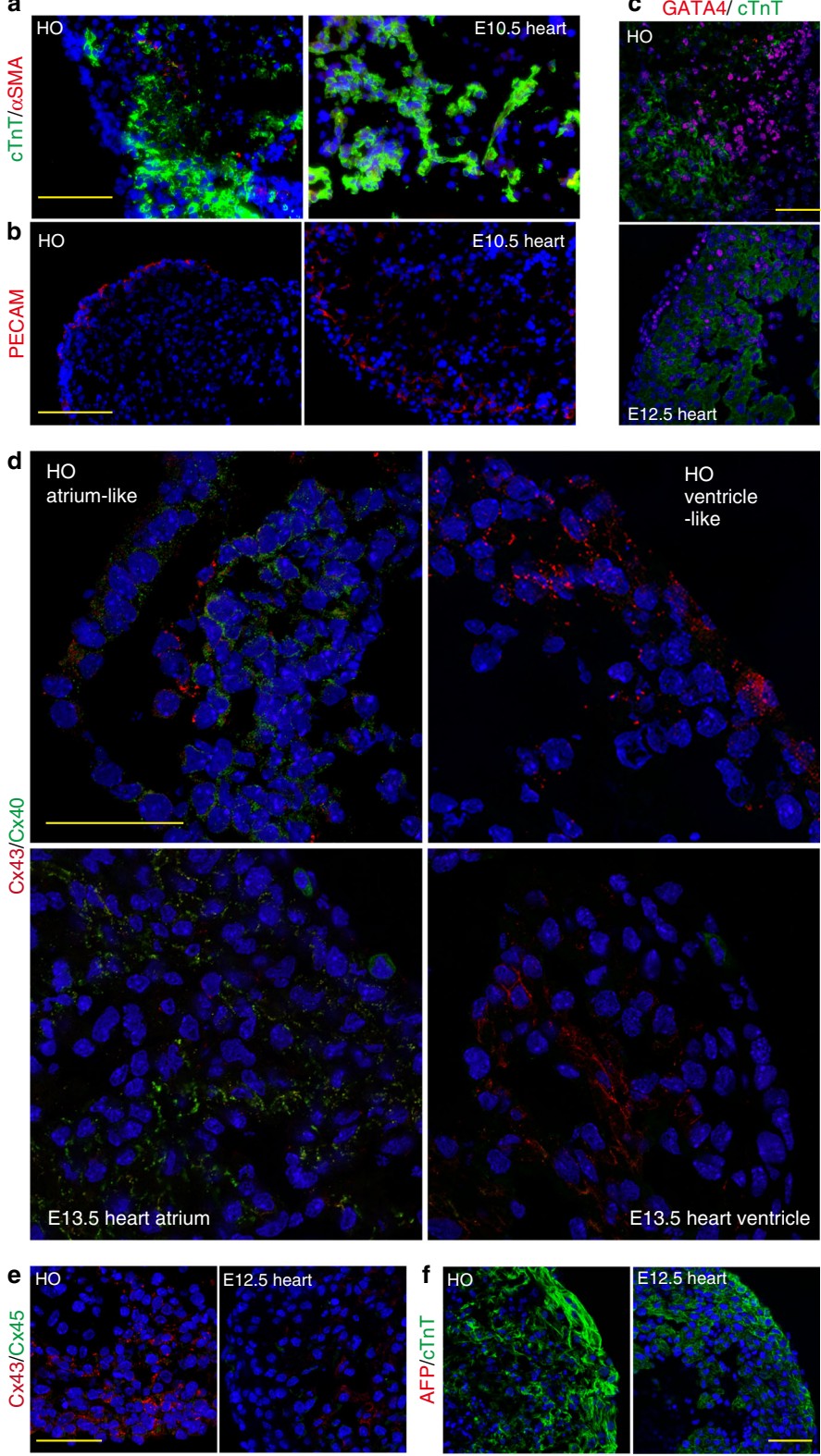

**Fig. 5 Evaluation of the cell components of heart organoid (HO)s. a** Immunofluorescent staining to detect cTnT (a CM marker, green)/αSMA (an SM marker, red) and **b** PECAM (an EC marker, red) in the HO and E10.5 heart cryosections. Expression of cTnT and αSMA was observed in the HOs (*n* = 7 out of *n* = 9, two experiments). Also, segregated lining expression of PECAM (red) was detected in the HOs (over *n* = 9, two experiments). Scale bar: 100 μm. **c–f** Immunostaining to detect **c** GATA4 (red)/cTnT (green), **d** Cx43 (red)/Cx40 (green), **e** Cx43 (red)/Cx45 (green), and **f** AFP (red)/cTnT (green). Scale bar: 50 μm. **c** Expression of GATA4, a cardiac TF, was observed in both HOs (*n* = 8 out of *n* = 9, two experiments) and E12.5 heart. **d** Expression of Cx40 in atrium-like region detected in the HOs (*n* = 7 out of *n* = 13, two experiments) was similar to that of atrium in E13.5 heart. **e** The organized distribution of two types of Connexin (Cx43 and Cx45) in the HOs (over *n* = 8, two experiments) was similar to that of E12.5 heart (the ventricle part), although a small number of Cx45 (green)-positive cells was observed. **f** No expression of AFP (a liver marker) was detected in both HOs (*n* = 8, over two biologically independent experiments) and E12.5 heart.

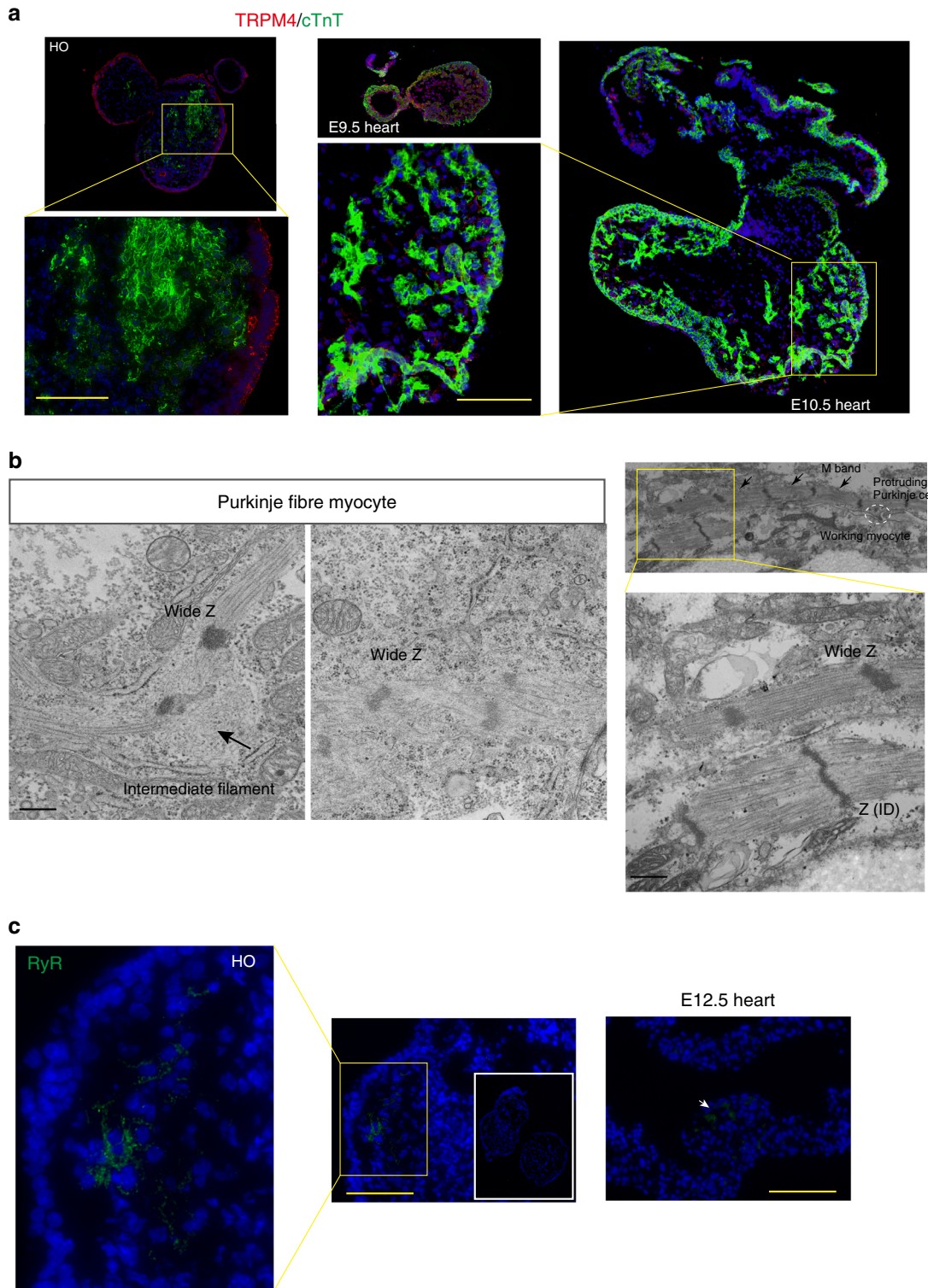

**Fig. 6 Purkinje cells and T-tubule-associated components in heart organoid (HO)s. a** Immunostaining to detect TRPM4 (red), a Purkinje cell marker, and cTnT (green) in HOs (left panels, *n* = 14, two independent experiments) and in E9.5 and E10.5 hearts (right panels). Scale bar: 100 μm. **b** Purkinje-specific myocytes in the HOs (over *n* = 3, over two experiments). A wide Z band, abundant intermediate filaments, and/or sparse myofibrils were observed (left panels). Both Purkinje cells (M bands (arrows) and wide Z bands) and working (ordinary) myocytes in the HOs were observed together in cell-to-cell contact, showing the protrusion of Purkinje cells (white dotted circle) into working myocytes (right panels). Scale bar: 400 nm. **c** RYR2 immunostaining in the HOs (left and middle panel, *n* = 13, three biologically independent experiments) and an E12.5 heart (right). Inset indicates an entire section of the HO. Scale bar: 100 μm.

Vesiculated nerve processes containing vesicles and glycogens make intimate contact with cardiac muscle cells[44]. We performed the ultrastructural analysis of heart organoids to determine whether cardiac autonomic nerves were formed in the heart organoids cultured in the presence of BMP4 from day 9. Notably, naked-neuronal axon bundles containing microtubules, small mitochondria, SR, and/or clear vesicles were found in the heart organoids cultured for 11 days (at least 9 out of 11 heart organoids, Supplementary Fig. 5b). Moreover, these neuronal axons were in tight contact with myocytes in the heart organoids (Supplementary Fig. 5b, left panel).

Based on these results, the heart organoids appeared to exhibit the organized structural properties of the embryonic hearts (CMs, SMs and ECs) as well as the cytological and histological maturity, such as the presence of Purkinje fibers, the association of SR/T tubule junction and the tight neuronal contact with the myocytes.

### Global gene expression patterns in the heart organoids.
To determine how closely the profile in the heart organoids matched that of embryonic hearts, we performed RNA-seq analysis of the EBs without further culturing ($n = 3$), of heart organoids cultured for 9 days ($n = 3$) and 13 days ($n = 3$), and of in vivo hearts from mouse embryos at E9.5 (referred to hereafter as E9.5H; $n = 2$) and E11.5 (E11.5H; $n = 3$). Principal component analysis (PCA) suggested that, as the heart organoids developed, their gene expression patterns transitioned from resembling that of the EBs to resembling that of embryonic hearts (Fig. 7a). Moreover, heat map and hierarchical clustering analyses based on the differentially expressed genes between the in vivo embryonic hearts and the EBs revealed that the gene expression in heart organoids was more similar to that observed in embryonic hearts at E9.5 and E11.5 than it was to that observed in EBs (Fig. 7b). Although the hemoglobin complex-related genes, including *Hbb-y*, *Hba-x*, and *Hbb-bh*, were expressed in in vivo embryonic hearts with blood, these genes were not expressed in the heart organoids (Fig. 7b; Supplementary Data 1). The gene expression profile of the representative genes in four of five categories demonstrated that the gene expression patterns in the heart organoids were similar to those of the in vivo embryonic hearts, including atrial/ventricle regional markers, markers during embryonic heart development, general stem cell markers and CM markers (Fig. 7c). Higher expressions of other tissue markers, such as *Foxa2/Sox17*, markers of the liver and intestine, and *Pdx1*, a marker of the pancreas in some of the heart organoids, may represent insufficient repression of developmental stage-specific markers during heart organoid generation because *Foxa2* is usually expressed in the EBs, *Sox17* also in the EBs and in endocardium of the heart during cardiogenesis[45], and repression of *Pdx1* was observed in the day 13 heart organoids (Fig. 7c).

To elucidate the features of differentially expressed genes between the heart organoids and the EBs, genes exhibiting a fivefold increase (396 genes with a log2-fold change > 5) or a fivefold decrease (55 genes with a log2-fold change < −5) were selected to create a log ratio mean average (MA) plot between the heart organoids and the EBs (Fig. 7d). Gene Ontology (GO) analysis of the 396 upregulated genes showed that 26 of the top 30 GO terms contained muscle contraction, heart process, heart contraction, cardiac muscle tissue development, circulatory system process, and sarcomere organization (Fig. 7e, top). By contrast, heart-related terms did not feature in any of the top 30 GO terms for the 55 downregulated genes (Fig. 7e, bottom).

Next, we investigated the expressions of coronary circulation-related genes in the heart organoids. The development of coronary circulation is heralded by the appearance of budding and canalized venous sprouts from the sinus venosus which contains vein cells undergo an early cell fate switch to create pre-artery population to build coronary arteries[4]. Especially, *Nr2f2* (*Coup-tf2*), a venous transcription factor, regulates the expression of artery genes. Notably, the gene expression level of the venous marker *Nr2f2* in the heart organoids was similar to that in in vivo embryonic hearts (Supplementary Fig. 5c, top). Similarly, in almost all of the heart organoids (regardless of the culture period), the gene expression patterns of the artery markers *Cxcr4*, *Jag2* and *Msx1* were similar to those of in vivo embryonic hearts from E9.5 to E12.5, although other artery markers, including *Notch4*, *Dll4* and *Gja4*, were only expressed at low levels in heart organoids cultured for 6 days (Supplementary Fig. 5c, bottom).

These results indicated that, overall, the heart organoid mimics the specific features of in vivo embryonic hearts in gene expressions, reflecting the diverse property of mammalian cardiac development including coronary circulation.

### Atrium- and ventricle-like structure in the heart organoids.
To investigate whether the heart organoids underwent chamber-like formation with regional expression of the CM subtypes (i.e., ventricular- and atrial-type CMs), we examined the expression of two sarcomere proteins in the heart organoids: myosin light chain 2 atrial (Mlc2a) and myosin light chain 2 ventricular (Mlc2v)[46]. Strong expression levels of Mlc2v were detected in almost all of the heart organoid cryosections ($n = 24$, Supplementary Fig. 6a) and segregated expression of Mlc2a was detected in most of cryosections of heart organoids (75%, $n = 24$, Supplementary Fig. 6a), whereas weak segregation in the remaining sections (25%, $n = 24$, Supplementary Fig. 6a). High-resolution images also displayed Mlc2a-positive individual cells in the atrium-like part and Mlc2v-positive individual cells in the ventricle-like part in the heart organoid (Fig. 8a). These results suggest that this culture system effectively induced CM formation, resembling that observed in vivo after E10.5. Meanwhile, in E10.5 embryonic hearts after chamber formation, Mlc2a was expressed in the atria, whereas Mlc2v expression was observed in the ventricles. Particularly strong expression of Mlc2v was observed in the LV of the E12.5 heart (Fig. 8a, right). Notably, whole-mount immunostaining combined with tissue clearing (optimized clear, unobstructed brain/body imaging cocktails and computational (CUBIC)) analysis[47] showed clear spatial separation of Mlc2a and Mlc2v expression in the heart organoids, suggesting successful generation of both atrium- and ventricle-like structures (Fig. 8b and Supplementary Movie 4).

In terms of ultrastructural features of atrial and ventricular myocytes, atrial myocytes are different from ventricular myocytes in both functional aspects, including atria-specific excitation and faster contractions, and morphological features, such as having more abundant organelles, prominent perinuclear Golgi complexes associated with atria-specific granules, numerous peripheral SR, and a high density of mitochondria dispersed in cells and secretory vesicles (SVs)[48]. In contrast to atrial myocytes, ventricular myocytes have well-structured sarcomeres with Z bands and myofibrils containing glycogens. Therefore, we further analyzed the ultrastructure of CMs in the heart organoids to confirm their atrium and ventricle-specific structures. Notably, cells exhibiting these features of atrial myocytes and ventricular myocytes, respectively, were located separately in the heart organoids (Fig. 8c), suggesting that this culture protocol promotes both of the atrial and ventricular formation process in vitro.

To further confirm the differential gene expression between the atrium- and ventricle-like regions in heart organoids, we performed RNA-seq analysis of the surgically dissected parts corresponding to these regions, and compared with the gene

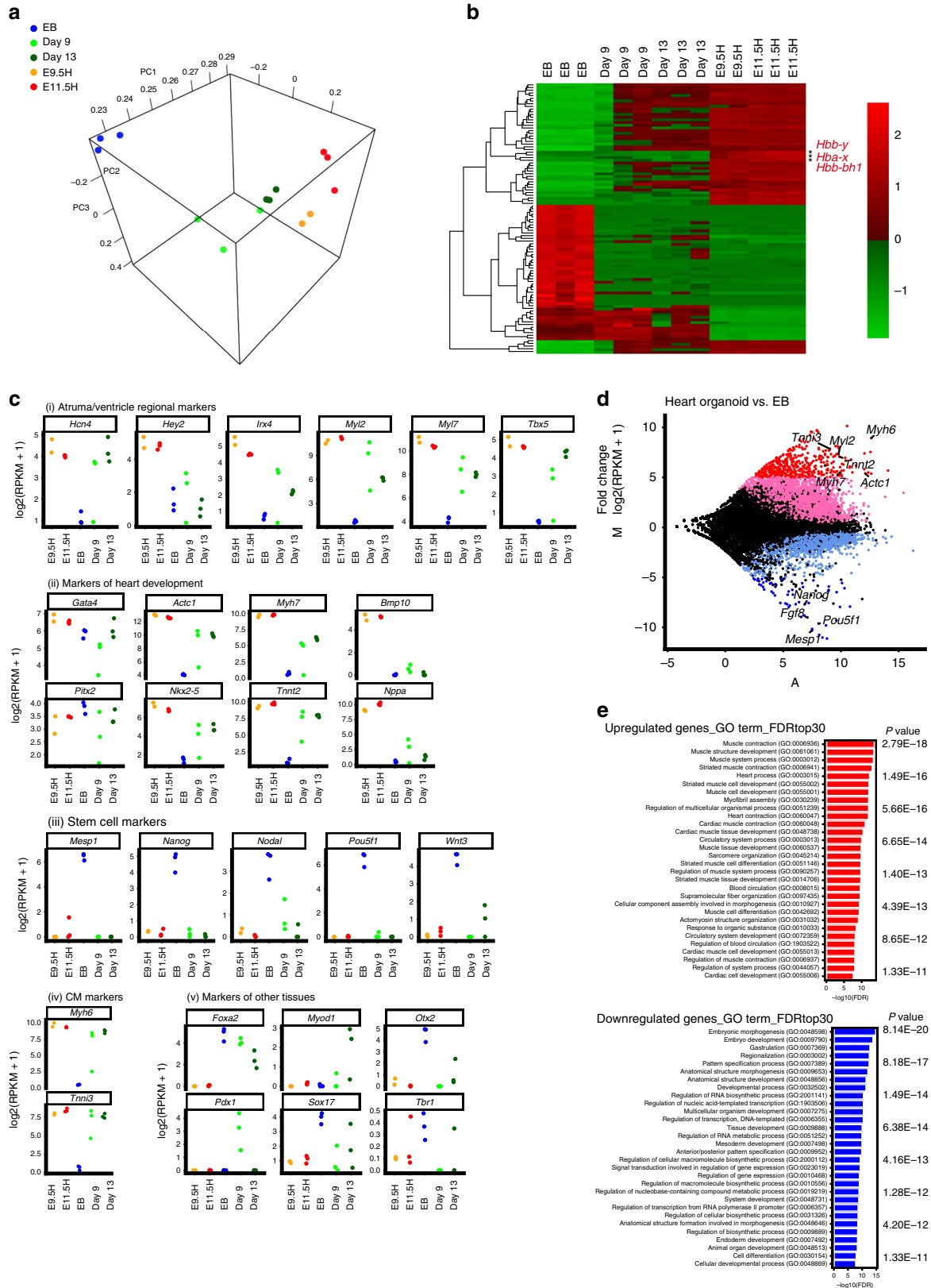

expression profiles of the heart organoids and embryonic hearts at day 11.5. Expression levels based on the differentially expressed genes (106 genes) between the atria and the ventricles of embryonic hearts revealed that the several gene expression patterns of atrium-like and ventricle-like regions within the heart organoids were similar to those observed in the embryonic hearts

at day 11.5; for example, *Col6a2, Gstm6, Slc7a7* showed the higher expressions in atria in vivo and atrium-like parts in heart organoids, and *Cend1, Gja1(Cx43), Sema3c, Cspg4, Lng1*, and *Snai3* showed the higher expressions in ventricles in vivo and ventricle-like parts in heart organoids, although *Bmp10* was not expressed in both atrium- and ventricle-like parts in heart

**Fig. 7 Global gene expression in heart organoid (HO)s. a** Principle component analysis of gene expression in HOs cultured for 9 days ($n = 3$) and for 13 days ($n = 3$), and in EBs ($n = 3$), E9.5 hearts ($n = 2$), and E11.5 hearts ($n = 3$). The HOs are located near the hearts. Hemoglobin complex-related genes, including *Hbb-y*, *Hba-x*, and *Hbb-bh*, were excluded from the PCA analysis. **b** Heat map and hierarchical clustering analyses based on the differentially expressed genes between the in vivo embryonic hearts (E9.5 and E11.5) and the EBs without further culturing. *Hbb-y*, *Hba-x*, and *Hbb-bh* are indicated with asterisks. **c** Gene expression profiles of representative genes for (i) atrium/ventricle regional markers, (ii) markers during embryonic heart development, (iii) general stem cell markers, (iv) CM markers and (v) markers for other tissues including *Foxa2/Sox17* (liver and intestine), *Pdx1* (pancreas), *Myod1* (skeletal muscle), and *Otx2/Tbr1* (central nervous system). **d** MA plot showing the differentially expressed genes between the HOs and EBs (the *y* axis indicates the log intensity ratio, and the *x* axis indicates the mean average of log intensity). Genes with a log2-fold change > 5 are shown in red (396 genes), and those with a log2-fold change < −5 are shown in blue (55 genes). **e** GO analysis based on the 396 upregulated genes of (**d**) ($P$ value = from 2.79E−18 to 3.89E−11; two-sided Wald test, log2-fold change > 5, middle panel) and the 55 downregulated genes of (**d**) (bottom; $P$ value = from 8.14E−20 to 2.21E−11; two-sided Wald test, log2-fold change < −5, bottom panel). Source data are provided as a source data file.

---

organoids (Fig. 8d and Supplementary Fig. 6b). However, the global gene expression profiles regarding the atrial/ventricular segregation in heart organoids are less robust as in E11.5H probably due to contaminating cells. In addition, we investigated the gene expression profile of classical ventricular- and atrial-specific genes in atrium- and ventricle-like parts in heart organoids based on the differentially expressed genes (951 genes with −1 < log2-fold change > 1, adjusted $P$ value < 0.05) between atria and ventricles of E11.5H. The ventricular-specific genes such as *Myl2* (*Mlc2v*), *Myl3* (*Mlc1v*), *Hand1*, and *Myh10* exhibited higher expression in ventricle-like part of heart organoids similar to that in ventricles of E11.5H (Fig. 8e, top), while the atrial-specific genes such as *Myl7* (*Mlc2a*), *Kcnj3*, *Tbx5*, and *Nr2f2* (also known as a venous transcription factor)[49] displayed lower expression in atrium-like part of heart organoids different to that in atria of E11.5H (Fig. 8e, bottom). This discrepancy in gene expression profiles at the atrium-like part may represent incomplete atrial specification in organoid culture presumably due to the decreased expressions of atrial TFs (*Tbx5* and *Nr2f2*) which promote expressions of many atrial-specific genes, or the insufficient surgical dissection of the heart organoids.

**Functional capacity of the cultured heart organoids.** As spontaneous beating was continuously observed in the heart organoids (Supplementary Movie 1; culture day 11), we next asked whether the heart organoids exhibited the $Ca^{2+}$ transients and cell excitability associated with beating in embryonic hearts. To assess the presence of $Ca^{2+}$ transients, $Ca^{2+}$ levels in intact 3D heart organoids cultured for longer than 9 days were analyzed using a $Ca^{2+}$-binding fluorescent dye. We identified transient increases in free $Ca^{2+}$ concentrations, indicating that the CMs in the heart organoids contracted with $Ca^{2+}$ transients (Fig. 9a and Supplementary Fig. 7, Supplementary Movie 5). In addition, administering isoproterenol, a β adrenergic agonist, led to the increase of beating rate and the decrease of time constant of $Ca^{2+}$ decay ($n = 6$, $P < 0.05$ by paired $t$ test, Fig. 9b), similar to the response to β adrenergic stimulation in vivo.

Next, we investigated the expression of the inwardly rectifying $K^+$ channel (IK1)/Kir2.1, which contributes to cell excitability and action potential regulation in the heart and is expressed at high levels in the late embryonic and postnatal heart[50,51]. Immunofluorescence staining revealed IK1 expression in the heart organoids (83%, $n = 10$ out of 12) similar to that of E12.5H and postnatal day 1 (P1) heart, indicating that the heart organoids were developmentally more mature than the early embryonic hearts (E9.5H and 10.5H without IK1 expression) in terms of IK1 (Fig. 9c). Next, to determine whether appropriate excitation occurs in the heart organoids, we performed an electrophysiological evaluation by optical mapping, which enables high-resolution analysis of membrane potentials[52]. Importantly, optical mapping using voltage-sensitive dye identified two distinct regions representing atrium- and ventricle-like activation such as

atrium-like propagation before ventricle activation (Supplementary Movie 6). These two components (atrium- and ventricle-like) of activation were detected in 13 of 22 heart organoids (59%). Activation of the atrium-like region preceded that of other area, and was characterized by relatively shorter action potential duration (APD). The ventricle-like region was excited after the atrium-like region with substantial conduction delay, and showed longer action potential duration (Fig. 10a). The atrium-like regions of different heart organoids exhibited significantly shorter APD90 than the ventricle-like region (Fig. 10b). These findings were similar to the observation of optical mapping of adult whole heart. Treatment with 100 nM of E4031, an IKr blocker, caused significant prolonged APD90 (Fig. 10c, left; graph, right; representative image). Moreover, treatment with 1 μM of E4031 led to spontaneous induction of tachyarrhythmia from an extrasystole (Fig. 10d, right and Supplementary Movie 7).

Taken together, these results suggest that the heart organoids generated using our approach mimic key features of $Ca^{2+}$ transient formation and cell excitability seen in embryonic hearts.

## Discussion

The present study demonstrated that employing FGF4 under serum-free conditions in a fetal-heart-fitted ECM environment generated the heart organoids could mimic the in vivo heart developmental processes in a spatial and temporal manner like self-organization. The protocol described herein enabled the generation of 3D heart organoids composed of CMs, ECs, SMs and potentially cardiac autonomic neurons. Importantly, we demonstrated that these 3D heart organoids exhibited structural and functional characteristics resembling their in vivo counterparts based on the presence of (1) cardiac muscle-specific IDs and T tubule/SR junctions in working CMs, (2) atrial myocytes with secretory vesicles as well as abundant other organelles, (3) Purkinje-specific myocytes suggesting the formation of a conduction system, (4) the expression of cardiac TFs and multiple structural proteins, and (5) functional $Ca^{2+}$ oscillations over a short culture period.

Notably, these heart organoids possessed atrium-like part with atrial myocytes at an ultrastructural level and ventricle-like part with ventricular myocytes. Thus, they mimic mammalian four-chambered heart formation in a semi-self-regulated manner, implying this process includes several essential gene networks and cell−cell interactions. It clearly demonstrates the critical importance of adequate ECM conditions to induce in vitro organ/tissue formation. It will be of great interest to assess the extent to which mimicking the basic mammalian body plan in vitro. Another interesting question is whether it is possible to trace the evolutionary path of vertebrate hearts, from fish (one atrium and one ventricle) to mammals (two atria and two ventricles) in vitro.

Also, the expression of Nestin (fetal heart and brain marker) was observed in both heart organoids and in E12.5 hearts,

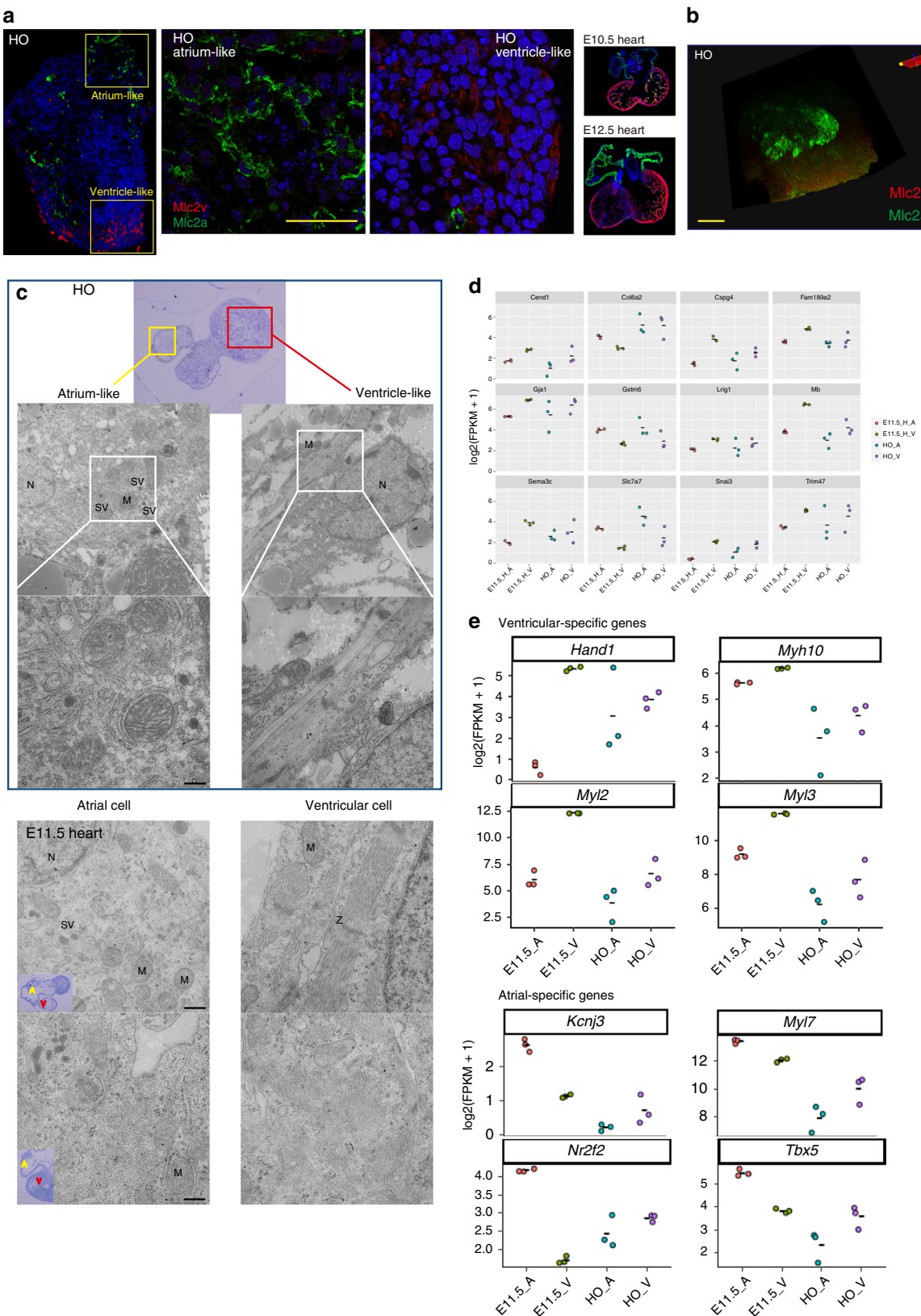

suggesting that Nestin-positive cells derived from NCCs may contribute to cardiac innervation and conduction through the cardiac autonomic nervous system[25,53].

Although the heart organoids generated with this protocol do not perfectly match the in vivo heart with respect to the global gene expression profile, the heart organoids did exhibit appropriate differentiation of various heart-specific components related to overall cardiac function. In particular, the heart organoids exhibited β adrenergic stimulation in response to isoproterenol treatment, demonstrating that these structures closely resemble the in vivo heart. In our heart-like generating system, administration of E4031 also causes prolongation of action

**Fig. 8 Formation of atrium- and ventricle-like structures in heart organoid (HO)s. a** Immunostaining for Mlc2v (red) and Mlc2a (green) in the HOs ($n =$ 30, over three independent experiments). Left: a whole HO. Middle: high-resolution images of the atrium- and ventricle-like regions corresponding to the insets in left panel. Right: embryonic hearts from E10.5 and E12.5. Scale bar: 50 µm. See also Supplementary Fig. 6a. **b** Whole-mount immunostaining of Mlc2v (red) and Mlc2a (green) in the HOs. Distinct Mlc2a-positive atrial cells were detected in the HO (over $n = 6$, over three independent experiments). Scale bar: 50 µm. **c** The HOs (over $n = 3$, over two experiments) possessed mainly two parts (top panel; a section stained with toluidine blue) with cells exhibiting features of the atrial myocytes (high density of mitochondria dispersed in cells and secretory vesicles, left) and features of the ventricular myocytes (well-structured sarcomeres with Z band and myofibrils containing glycogens, right). These morphological features are similar to those of the in vivo embryonic hearts. M mitochondria, SV secretory vesicles, G Golgi complex, N nucleus, RER ribosomal endoplasmic reticulum. A (yellow), atrium; V (red), ventricle in the insets. Scale bar: 400 nm. **d** Gene expression pattern of atrium- and ventricle-like structures in HOs. The HOs cultured for 13 days ($n = 3$) were surgically dissected to each atrium- and ventricle-like parts for RNA-seq analysis. Specific genes showing the similar chamber expression patterns in HOs and E11.5H. See also Supplementary Fig. 6b. Source data are provided as a source data file. **e** Ventricular- and atrial-specific gene expression profiles in dissected HOs and E11.5H based on the 951 differentially expressed genes ($-1 < \log2$-fold change $> 1$, adjusted $P$ value $< 0.05$) between atria and ventricles of E11.5H.

potential duration and tachyarrhythmia through optical mapping analysis like the effect of E4031 in normal CMs, reflecting certain features of mutations in IKr channel genes such as hERG and KCNH2. Moreover, the heart organoids demonstrated autonomous beating with myocardial contractions, thereby fulfilling the criteria for a functional heart.

Thus, our heart organoid culture system provides an efficient model to study the pathologies of congenital heart defects with structural abnormalities and to screen potentially dangerous drugs that cause cardiac defects, possibly leading to significant progress in regenerative medicine.

Another notable result is the expression of IK1 and RYR2 in the heart organoids. IK1 expression has not been detected in previous iPSC-derived CMs, even though the presence of the K+ channel is a key criterion for mature CMs, which are used to assess heart toxicology. Given that our electrophysiological assessment revealed normal excitement in the heart organoids and altered excitement in the heart organoids treated with the IKr blocker E4031, this culture system might be suitable for use in screening for drug side effects such as arrhythmia. Thus, we expect that this method will be used to generate 3D heart organoids from human iPSCs for translational research applications. A few studies[54–56] have recently reported the successful bioengineering of cardiac muscle tissue from human iPS-derived CMs, and these platforms will be a promising tool for drug testing. Our culturing methods also recapitulate cardiac perturbation in vitro by the deletion of cardiac TF genes such as *Tbx5* and *Isl1* important for heart development. As our culturing method for heart organoids containing cardiac muscle and other components is a unique biomimetic model for embryonic heart development in a self-organization-like manner from pluripotent stem cells, this system will be useful for evaluating the heart toxicity of newly developed drugs in the future.

## Methods

**Animals**. All animal experiments were approved by the Institutional Animal Care and Use Committee of Tokyo Medical and Dental University (TMDU), and were performed according to the guidelines of the same institution. Mice were allowed access to a standard chow diet and water ad libitum and were housed in a pathogen-free barrier facility with a 12L:12D cycle, at temperature and humidity ranges of 22−24 °C and 40−60%, respectively.

**Cell culture and heart organoid generation**. A step-by-step protocol describing the heart organoid generation protocol can be found at Protocol Exchange[57]. Trypsinized wild-type ESCs (three independent cell lines derived from C57BL/6 and F1 hybrid C57BL/6× JF1) were seeded into six-well plates coated with 0.2% gelatin and incubated at 37 °C for 45 min. Floating ESCs were collected and centrifuged at 1000 rpm for 5 min[17]. For EB formation, 1000−5000 ESCs (or 500 ESCs, for the examination of the minimum number of ESCs required for heart organoid generation) were seeded onto each well of a U-bottomed 96-well plate (Prime surface 96-well U, Sumiron) in FBS-supplemented ES medium without LIF and incubated at 37 °C for 4 days. For heart organoid cultures, EBs were transferred onto an 85.7 µg/cm² LN/ET gel (BD 354259) or onto 10.7 µg/cm² iMatrix-411 (gifted from Nippi)-coated chamber slides (Falcon 35418) in 200 µl of heart

organoid medium (50 µg/ml penicillin/streptomycin, 20% KSR, 1 mM sodium pyruvate, 100 µM β-mercaptoethanol, 2 mM L-glutamine, 60 ng/ml progesterone, 30 ng/ml β-estradiol, 5 µg/ml insulin, 20 µg/ml transferrin, and 30 nM selenite in DMEM/F12 (Gibco 11320033)) containing FGF4 (30−60 ng/ml). Subsequently, the EBs were incubated at 37 °C and 5% $CO_2$ for 10−15 days or longer with medium changes on days 3, 5, 7, 9, 11, 13, 14, and 15. Starting on day 9, BMP4 (50 ng/ml), BIO (2.5 µM), and LIF (1000 units/ml)[14] were added to the medium. After culturing, the heart organoids were collected for further analysis. In case of other growth factors, 30 ng/ml of bFGF (FGF2) and 60 ng/ml of FGF10 were used. For Matrigel culture, EBs were embedded in Matrigel (9−12 mg/ml, BD356230) drops.

**Establishment of Tbx5 KO and Isl1 KO ESC lines using the CRISPR/Cas9 system**. For the generation of *Tbx5* KO vector, sgRNAs were designed with the CRISPR design tool, Benchling (www.benchling.com). In case of *Isl1*, sgRNAs were used as described previously[33]. The used sequences of sgRNAs are as follows: *Tbx5* sgRNA1: 5′CGAAACCTGAGAGTGCTCTG, *Tbx5* sgRNA2: 5′CAAGTCTC CATCATCCCCGC, *Isl1* sgRNA1: 5′CCGATTTAAGCCGGCGGAGT, and *Isl1* sgRNA2: 5′TCATGAGCGCATCTGGCCGA. Each sgRNA was annealed and inserted into the pGuide-it-ZsGreen1 Vector (Guide-it CRISPR/Cas9 system (Green), Clontech) as per the manufacturer's protocol. Both of pGuide-it-ZsGreen1-sgRNA1 and -sgRNA2 for each gene were transfected to wild-type (WT) B6 ES cells by electroporation using Super electroporator NEPA 21 typeII (NEPA GENE). To subclone KO cell lines, GFP-positive ES colonies were randomly picked and propagated. All subclones were investigated for their mutations using Guide-it mutation detection kit (Clontech). Six lines of *Tbx5* KO ESC lines and six lines of *Isl1* KO ESC lines were established and each of the three KO ESC lines were used for the generation of heart organoids with FGF4 on LN/ET complex.

**RNA-seq analysis**. EBs without further culturing and heart organoids cultured for 6 ($n = 3$), 9 ($n = 3$), 13 ($n = 3$), or 15 (large, $n = 3$; small, $n = 3$) days were collected and immediately frozen in liquid nitrogen. In addition, dissected embryonic hearts at days 9.5 and 11.5 (two or three biological replicates) were collected and frozen as described above. Total RNA was extracted using an AllPrep DNA/RNA Micro Kit (QIAGEN) according to the manufacturer's protocol. Libraries for the RNA-seq analysis were prepared using a KAPA Stranded mRNA-Seq Kit (KAPA Biosystems). The resulting libraries were sequenced on a HiSeq 1500 (single-end 50-bp reads with HiSeq SR Rapid Cluster Kit v2 and HiSeq Rapid SBS Kit v2, Illumina). The sequence data were mapped against the mouse reference genome sequence (GRCm38/mm10) using Bowtie2, followed by calculation of the read numbers and RPKM value for each gene using the Bioconductor package DEGseq. For the RNA-seq analysis of atrium- and ventricle-like structures in heart organoids, 13-day cultured heart organoids ($n = 3$, biologically triplicates) were surgically dissected to each atrium-like part and ventricle-like part and immediately frozen in liquid nitrogen. Also, embryonic hearts at day 11.5 were surgically dissected to atria and ventricles. Total RNA was extracted as described above and the preparation of libraries and sequencing were performed by Takara company. The sequence data were mapped against the mouse reference genome sequence (GRCm38/mm10) using STAR, followed by calculation of the fragment numbers and FPKM (fragments per kilobase of exon per million mapped fragments) value for each gene.

**Immunofluorescent staining**. Heart organoids were collected after 10−15 days of culture and frozen in optimal cutting temperature (OCT) compound (Tissue-Tek) in a plastic tissue mold (Cryo dish, Shoei). The sections were sliced at a thickness of 5−7 µm using a cryostat set at −16 °C and transferred to MAS-coated glass slides (Matsunami) or poly-L-lysine-coated slides (Sigma). EBs without further culturing were also collected, frozen in OCT, and sectioned as described above.

To prepare mouse embryonic and postnatal heart sections, pregnant female mice (C57BL/6, from 9.5 to 13.5 days postcoitum) and postnatal day 1 (P1) mice were sacrificed, and the hearts were dissected from the embryos and neonates at each developmental stage. The embryonic hearts and P1 hearts were immersed in

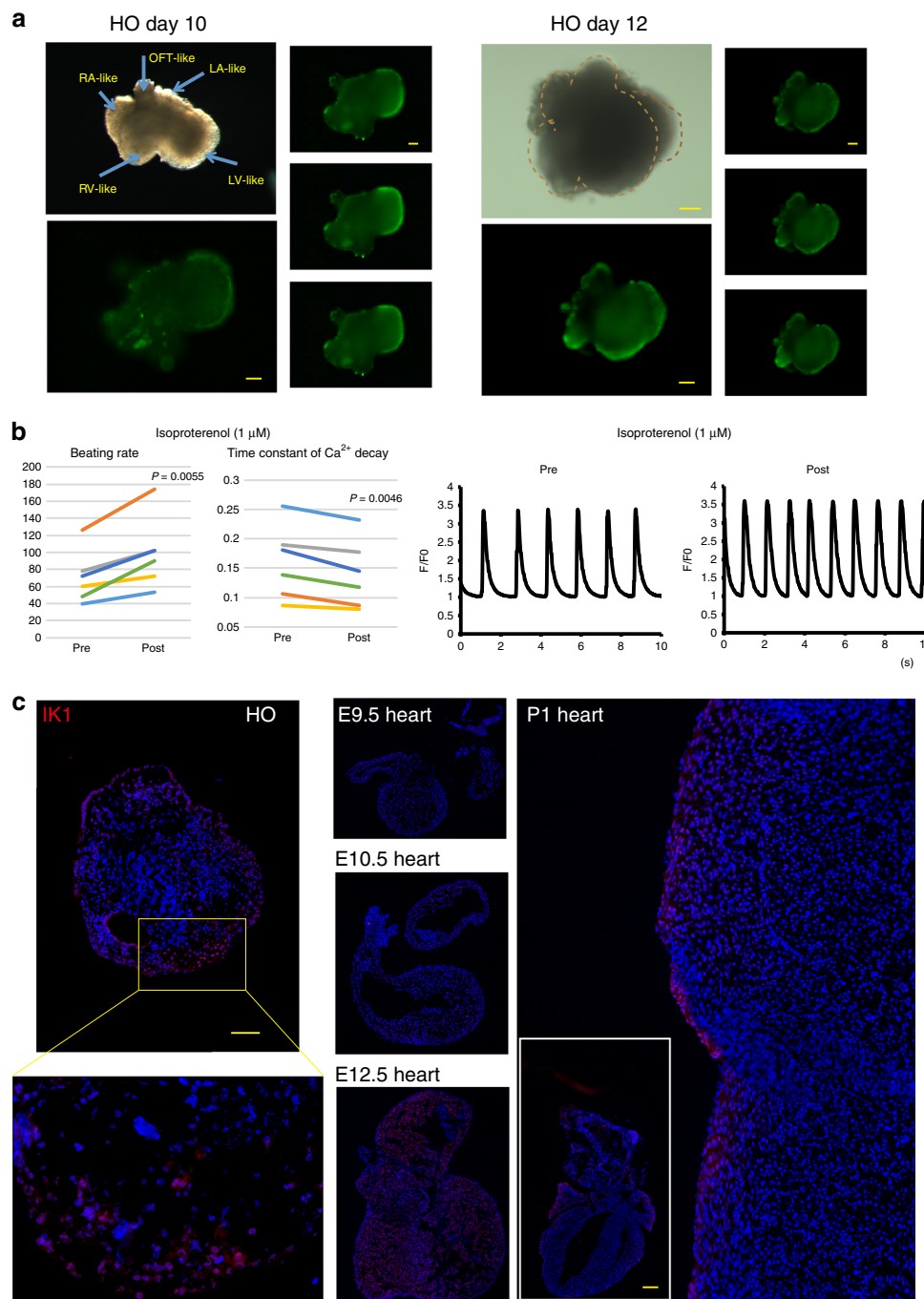

**Fig. 9 Functional maturity of heart organoid (HO)s. a** Ca$^{2+}$ transients in the HOs. Ca$^{2+}$ oscillations were analyzed using the Ca$^{2+}$-binding fluorescent dyes Fluo8 AM. Representative images of HOs ($n = 7$, two experiments, each top left panels, bright field and bottom left panels: fluorescent images). Three fluorescent images were extracted from every ten frames of sequential images (each right column, see also Supplementary Fig. 7). Scale bar: 100 μm. **b** Ca$^{2+}$ transients in the 11-day cultured HOs, pre- and post administration of the β adrenergic agonist isoproterenol (1 μM). Beating rate was increased and time constant of Ca$^{2+}$ decay was decreased after treatment of isoproterenol (left, $n = 6$, $P = 0.0055$ and 0.0046 by two-sided paired $t$ test). The representative traces of changed Ca$^{2+}$ signal (F/F0) before and after the treatment of isoproterenol (right). **c** IK1 immunostaining in a representative HO (left, $n = 10$ out of $n = 12$, two independent experiments), embryonic hearts (E9.5, E10.5 and E12.5; middle) and a postnatal day 1 (P1) mouse heart (right). Inset indicates the whole image of P1 mouse heart. Scale bar: 100 μm (HO), and 1 mm (P1 mouse heart).

sucrose-PBS solutions (4, 10, 15, or 20%) and then frozen in OCT, and later sliced at a thickness of 7 μm using a cryostat. The cryosections were fixed in 4% paraformaldehyde-PBS at room temperature (RT) for 15 min. The fixed sections were washed three times in PBS for 5 min and incubated with a blocking buffer (PBS containing 5% normal goat serum and 0.3% Triton X-100, or PBS containing 5% bovine serum albumin (BSA) and 0.3% Triton X-100) at RT for 1 h. The sections were subsequently incubated overnight with primary antibodies in dilution buffer (PBS containing 1% BSA and 0.3% Triton X-100) at 4 °C. The sections were then washed with PBS and incubated with secondary antibodies at RT for 1−2 h. The slides were counterstained with DAPI (diluted 1:1,000, Dojindo Laboratories) and mounted using a VECTASHIELD HardSet Antifade Kit (VECTOR Laboratories). Primary antibodies and dilutions used: anti-Tbx5 (1:100, Abcam, ab137833), anti-cTnI (1:50, ab47003), anti-cTnT (1:50−1:100, ab8295), anti-Nkx2-5 (1:50−1:100, ab91196), anti-Nestin (1:100, ab105389), anti-Oct3/4 (1:100, Santa Cruz Biotech, sc-8629), anti-PECAM

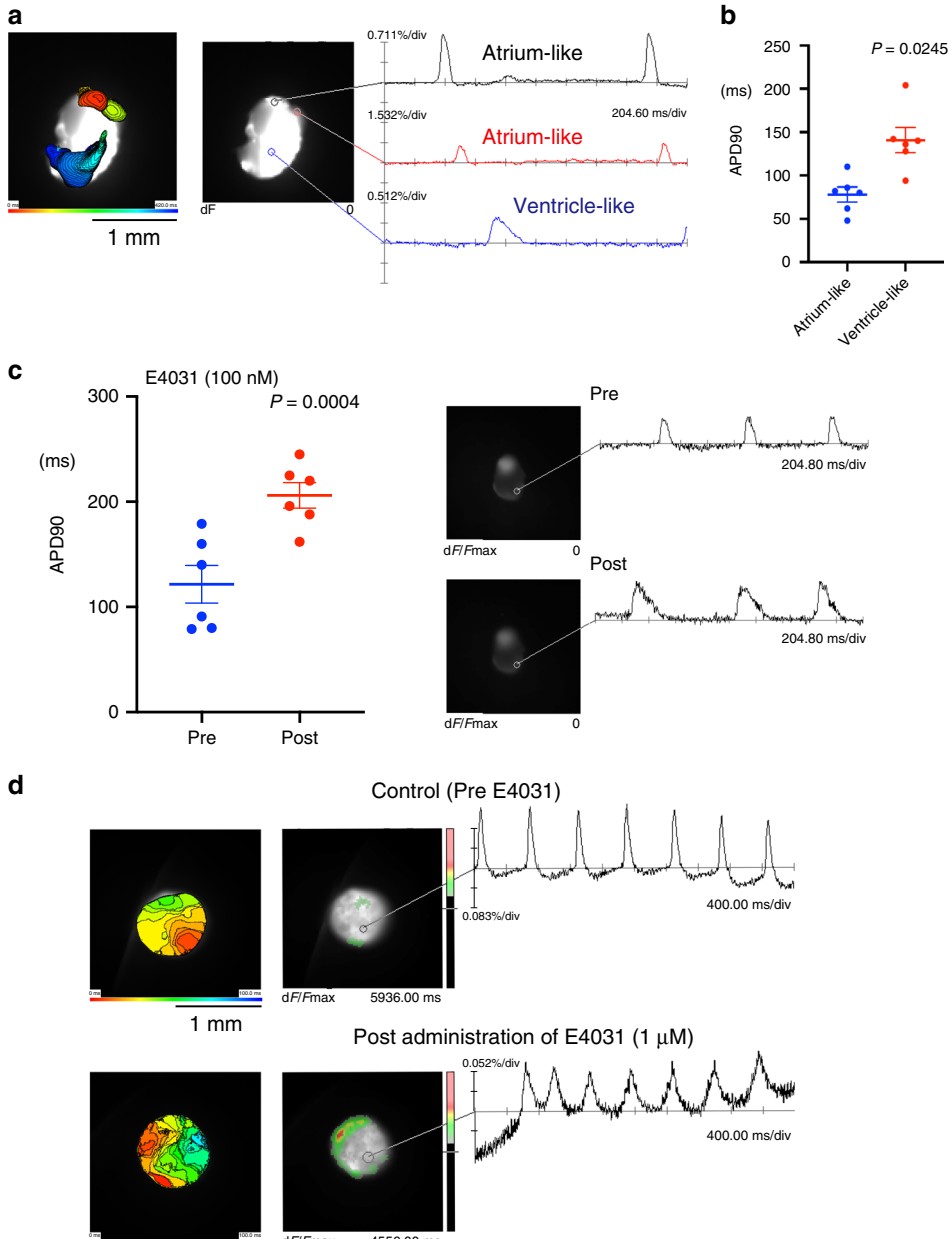

**Fig. 10 Electrophysiological properties of heart organoid (HO)s. a** Activation map during spontaneous excitation (left). Each local action potential of atrium-like (black and red) and ventricle-like (blue) components was presented in diagrams (right). **b** Action potential duration (APD90) in each atrium- and ventricle-like action potential of HOs ($n = 6$, $P = 0.0245$ by two-sided paired $t$ test). Data are presented as mean values ± SEM. **c** APD90 of HOs was analyzed by optical mapping in HOs pre- and post treatment with 100 nM of E4031 (left, $n = 6$, $P = 0.0004$ by two-sided paired $t$ test). Data are presented as mean values ± SEM. The representative optical mapping data of HOs pre- and post treatment with 100 nM of E4031 (right). **d** Action potentials recorded by optical mapping in HOs, pre- (top) and post (bottom) treatment with the IKr blocker E4031 (1 μM). High-resolution activation maps (left panels) show that the excitation pattern and excitation propagation speed are different in HOs before and after treatment with E4031 (right diagrams, see also Supplementary Movie 7).

(1:50, BD, #550274), anti-Mlc2a (1:50, Synaptic System #311 011), anti-Mlc2v (1:100, Synaptic System #310 003), anti-SM-MHC (1:50, R&D Systems, MAB4470), anti-αSMA (1:50, ab5694), anti-TRPM4 (1:50, ABN418), anti-KCNN4 (IK1, 1:50, GTX54786), anti-GATA4 (1:50, ab134057), anti-AFP (1:50, ab213328), anti-Cx43 (1:50, Sigma, C6219), anti-Cx40 (1:50, Invitrogen, #37-8900), anti-Cx45 (1:50, ab78408) and anti-RYR (1:100, ab2868). Secondary antibodies and dilutions used: Alexa 488 goat anti-mouse IgG1 (1:500, Invitrogen, A21121), Alexa 568 goat anti-rabbit IgG(H + L) (1:500, Invitrogen, A11036), Alexa 568 goat anti-rat IgG(H + L) (1:500, Invitrogen, A11077), Alexa 488 goat anti-rabbit IgG (1:500, Invitrogen, A11034), Goat anti-mouse IgG2b Secondary Ab Alexa Fluor 647 (1:500, Invitrogen, A21242), Alexa Fluor 488 goat anti-mouse IgG Ab (1:500, Invitrogen, A11001), Alexa

594 donkey anti-goat IgG(H + L) (1:500, Invitrogen, A11058), Alexa 488 donkey anti-rabbit IgG(H + L) (1:500, Invitrogen, A32790).

**3D imaging by optimized CUBIC tissue clearing.** For whole-mount immuno-fluorescence of cleared samples, we performed an optimized CUBIC analysis[47]. The collected heart organoids cultured for 11 days were fixed in 4% PFA/PBS overnight at 4 °C, incubated in Reagent 1 (25 wt% urea, 25 wt% N, N, N′, N′- Tetrakis (2-hydroxy propyl) ethylenediamine, 15 wt% Triton X-100, and 35 wt% DDW) in a shaker for 4 h at 37 °C and 80 rpm, and then washed with PBS three times for 20 min each. For immunostaining, the samples were incubated in a primary

antibody solution (1 wt% BSA, 0.02 wt% sodium azide, 10 vol% normal goat serum and 0.1 wt% Triton X-100 in PBS) overnight at 4 °C, washed with PBS three times for 20 min each, and incubated in a secondary antibody solution (0.2 wt% BSA, 0.02 wt% sodium azide, and 10 vol% normal goat serum in PBS) in a shaker for 12 h at 37 °C and 80 rpm. After the samples were washed with PBS three times for 10 min each, they were dehydrated with 20% sucrose in PBS for 30 min and finally incubated in reagent 2 (10 wt% triethanolamine, 50 wt% sucrose, 25 wt% urea, and 15 wt% DDW) in a shaker for over 2 h at 37 °C and 80 rpm. The immunostained heart organoids were analyzed with a laser confocal microscope (LSM 710, ZEISS, or FV3000, OLYMPUS).

**Ca$^{2+}$ measurement**. The heart organoids were washed two times with PBS, incubated at 37 °C for 15−30 min with 4 μM of the green fluorescent intracellular Ca$^{2+}$-binding dye Fluo8-AM (AAT Bioquest, Sunnyvale, CA) with Pluronic F127 (Dojindo, Tokyo, Japan) in normal Tyrode's solution, and then washed again two times with PBS. Subsequently, the heart organoids were mounted in glass-bottom dishes with 200 μl of normal Tyrode's solution and observed under a fluorescence microscope (BZ-X710, Keyence). Isoproterenol was used in the final concentration of 1 μM. The obtained images were analyzed with ImageJ software. The time constant of Ca$^{2+}$ decay was calculated by a single exponential fit using Origin software (OriginLab).

**Optical mapping**. Optical mapping was performed with a high-speed CMOS camera system (MiCAM Ultima, Brainvision, Tokyo, Japan) as reported previously[52]. The heart organoids were stained with 15 μM di-4-ANEPPS (Wako, Tokyo, Japan) for 15 min, followed by washout with PBS. Next, the samples were incubated with 30 μM of blebbistatin (Sigma-Aldrich, St. Louis, MO) for 15 min to eliminate the motion artifact. After washout, the heart organoids were placed in a glass-bottom dish filled with Tyrode's solution (135 mM NaCl, 5.4 mM KCl, 1.8 mM CaCl$_2$, 0.53 mM MgCl$_2$, 0.33 mM NaH$_2$PO$_4$, 5.5 mM D-glucose, and 5.0 mM HEPES at pH = 7.40 adjusted with NaOH and aerated with 100% O$_2$). E-4031 was used in the final concentration of 100 nM or 1 μM. The temperature was maintained at 37 °C throughout the procedure. All optical mapping was recorded using a ×5 objective lens, resulting in a spatial resolution of 20 μm × 20 μm/pixel, and data sampling was performed at 500 frames per second. The obtained data were analyzed using BV analysis software (Brainvision).

**Reporting summary**. Further information on research design is available in the Nature Research Reporting Summary linked to this article.

## Data availability

The raw sequencing data from the RNA-seq analyses have been deposited in the GEO database under accession code: GSE143932 (GEO) and under accession codes: SAMD00202559−SAMD00202583 (DDBJ BioSample). All relevant data are available from the authors upon reasonable request. Source data are provided with this paper.

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

## Acknowledgements

The authors would like to thank Drs. A. Kimura (TMDU), M. Watanabe (TMDU) and S. Nishikawa (Kyoto University) for helpful discussions, and the Research Core Center (RCC at TMDU) for the ultrathin section preparations used in the electron microscopy studies. The authors would like to thank Dr. Hayakawa (Sony) for performing the motion vector prediction analysis. This work was supported by the TMDU president grant to F.I. and Grant-in-Aid for Scientific Research (C) (20K06652) to J.L.

## Author contributions

J.L., A.S., R.K., T.S., J.T., T.K., K.I., K.T. and M.Y. performed the experiments and analyzed the data. J.T., T.M. and T.F. gave technical support and conceptual advice. J.L. and F.I. designed the study and wrote the manuscript.

## Competing interests

TMDU has filed a patent application (JP2017-190950, PCT/JP2018/36538, 18860752.7, 16/644,677 (US); ORGANOID AND METHOD FOR PRODUCING SAME) associated with the content of the manuscript. F.I. and J.L. are the inventors listed on the patent. All other authors declare no competing interests.
