## [Peer Review File · Nature Communications]

Reviewers' Comments:

Reviewer #1:

Remarks to the Author:

The manuscript entitled "In vitro recapitulation of embryonic heart structure and function via FGF4 and extracellular matrix" by Lee et al. describes a methodology resulting in the generation of cardiac organoid like structures from mouse embryonic stem cells. The authors used a combination of ECM modulation using laminin-entactin in conjunction with activation of FGF-4 to form a self-organizing organoid. The authors demonstrated the presence of chamber-like formation, chamber-specific gene expression, ultrastructural maturation, and the development of conduction tissues. They also identified the development of innervation and vascularization within the organoid, and demonstrated basic electrophysiological features of the generated organoids.

The generation of organoids that recapitulate embryonic cardiac development using an in-vitro system may be of great significance and with several applications ranging from basic developmental questions to assessment for drug toxicity and teratogenicity screening. Nevertheless, while the current study may be of great interest there are several key issues that need further evaluation to strengthen the scientific findings:

1. The reproducibility of the system is not clearly assessed throughout the manuscript. If this method is indeed effective and reproducible for each finding, the authors should clearly state the number of heart organoids (HOs) demonstrating the presented phenotype. For example, what percentage of HOs exhibited chamber-specific gene expression profile? What percentage had markers of cardiomyocyte maturation and sub-specification? How many included TRPM-4 expression? Innervation? Were their variation in the sizes of the different chamber-like structures?
2. The global HO gene expression analysis has nicely shown that HOs are more similar to the embryonic hearts than to EBs. Nevertheless, to further support the formation of different chamber types, regional (spatial) transcriptomics is required to ascertain chamber-specific gene expression patterns and reproducible and robust organoid self-organization. Thus the authors should use laser-dissection or other methods to perform detailed gene-expression analysis of the different regions within the organoids (for example, from the regions identified in Fig.1E).
3. In a similar manner, the investigators should use antibodies targeting multiple chamber-specific markers to further support the generation of distinct chambers.
4. To ascertain that the presumed HO morphologies are identifiable and reproducible, embryological perturbations are required. Silencing or knockout of genetic components or pharmacological inhibition of signaling pathways associated with chamber formation and/or cardiac looping (e.g. FGF-8) and/or primary and secondary field heart formation (e.g. ISL-1 knockout) should be assessed to demonstrate the ability of the in-vitro model to recapitulate known perturbations from the in-vivo environment.
5. Do the organoid display the presence of endocardial and epicardial cell-layers and if yes, do they express the necessary properties of such cell types?
6. The authors show that gene expression associated with endodermal structures (liver, intestine, pancreas) are expressed in the HOs (in higher levels than in in-vivo embryonic hearts). A purity assessment is required to ascertain that the presence of non-cardiac cells is indeed negligible, this should be done using both immunostaining, FACS for derivatives of non-mesodermal germ layers and single cell gene expression analysis.
7. The electrophysiological assessment does not assess the electrophysiological properties of the organoid and rather demonstrate characteristic cardiac tissue properties (partial isoproterenol response- chronotropic but not inotropic and APD prolongation). It is recommended that the EP

evaluation will focus on the differential properties of atrial/ ventricular activation. Presence of components of the conduction system (SA node, AV node), conduction velocity in and between the formed chambers and presence of conduction blocks at the different stages. For example SA node cells can be identified as NKX negative/troponin positive cells,

8. Connexin staining should be performed to evaluate the presence of gap junctions and connexin types (Cx43, Cx40, Cx45) since the type of connexin may correlate with the different cell/tissue types (atrial, ventricular, and Purkinje cells).

9. While the findings of the study is interesting in generating a organoid model with primitive chamber-like structures the discussion should be toned down since the structures derived are still far from recapitulating the in vivo heart.

10. Some of the immunostainings are of low resolution or are only provided in low-magnification and do not allow to identify cellular interactions. For example, it is not clear whether vessel like structure are forms.

Reviewer #2:

Remarks to the Author:

In the study, Lee et al. used mouse embryonic stem cells to established an in vitro heart organoid (HO) system, which is claimed to represent a "faithful biomimetic model of the developing heart". Generation of the HOs mainly followed established procedures but differs from previously described approaches by the addition of FGF4 to the culture medium and the use of commercially available LN/ET (laminin/entactin) gel. The authors observed the generation of different cardiac cell types (including atrial and ventricular myocytes, Purkinje fibers, endothelial and smooth muscle cells as well as neurons) and the formation of cystic structures, which seem to promote separation of atrial and ventricular myocytes. The authors performed various morphological, molecular (RNA-seq) electrophysiological measurements to analyze properties of HOs and characterize the state of cells within HOs compared to corresponding developmental stages during mouse embryonic development.

Recapitulation of the development of composite organs in vitro by mere self-organization is a very ambitious endeavor and probably not realistic when it comes to complex morphogenetic processes requiring the input of different cues (including mechanical forces and blood flow) provided by neighboring tissues and signaling centers. Generation of cystic structures that vaguely remind of cardiac chambers and contain different cardiac cell types is simply not enough to justify the authors' bold claim to recapitulate "embryonic heart structure and function". The authors make numerous additional statements, which are vastly exaggerated and not really supported by their findings, although the authors made some interesting observations that might move the field forward. The only observation, which I indeed found remarkable, was the "spontaneous" separation of what seem to be atrial and ventricular cardiomyocytes in the HO cysts, which I have not seen published before.

Specific remarks

The authors have used FGF4 and commercial LN/ET gels to generate HO cysts but the use of these ingredients is poorly justified and badly controlled. I am not aware of a specific function of FGF4 during heart development. Why did the authors use FGF4 and not another FGF? Do other FGFs (e.g. FGF2, 6, 10) have the same properties as FGF4 for HO formation? In fact, several other groups have used FGFs to promote differentiation of ES cell to cardiomyocytes. For example, Chun et al. (PlosOne 2010) used FGF10 to facilitate cardiomyocyte differentiation and also the FGFR inhibitor PD173074 (same as used in this study) to prevent cardiomyocyte differentiation. Hence, it is hardly unexpected that inhibition of FGFR signaling prevents cardiomyocyte formation. I found

it also confusing that the authors sometimes refer to FGF4 as a factor that promotes proliferation of cardiomyocytes and sometimes as a factor that restricts proliferation and promotes differentiation.

The authors show that LN/ET gels provide an advantage for HO formation compared to coated chamber slides but no other comparisons were made. What happens when alternative ECM matrixes are used? How pure are the commercial LN/ET gels? Are there other components included? A more systematic evaluation of different matrix components would be very helpful to precisely define conditions favoring organoid formation.

It requires a lot of imagination to recognize the different stages of heart development (cardiac crescent, primary heart tube, turning, four-chambered heart) in the cystic structures presented by the authors. Unfortunately, I am lacking such skills. In Figure 1e, the authors labeled different areas of a poly-cyst as right/left ventricles and left/right atria. Why that? Only because some cells lack Tbx5 expression? A much more comprehensive characterization of the different parts of the organoids is required to arrive at such conclusions.

In Figure 4b, the authors claim to see trabeculation of HOs. I can hardly detect anything in Fig 4b. Specific markers for trabeculation are available, which need to be used before making such claims.

The authors completely ignored the role of primary and secondary heart fields contributing to cardiac growth and morphogenesis, which was nicely addressed in a similar organoid study published in NC last year. Do the different heart fields properly contribute to cardiac development of HOs as in vivo?

The authors claim that the current HO model is an excellent model for drug testing and for assessing the role of gene mutations for cardiac morphogenesis. Why didn't the authors simply used available mutant mouse ESCs (e.g. Isl1, Tbx5 or other) to substantiate the claim?

The separation of atrial and ventricular cardiomyocytes in the HOs is interesting and the authors did a good job investigating ultrastructural features of atrial and ventricular cardiomyocytes. However, it is clearly required to subject the potential "atrium" and "ventricle" structures to a more comprehensive characterization by RNA-seq. If the structures are really as distinct as claimed by the authors, it should not be problem to dissect them and sequence the different "chambers" separately.

In this context, I was also concerned that the authors wrote that weak and segregated expression of MLC2a was only visible in SOME HOs. In how many HOs did the authors observe strong and segregated expression of both MLC2a and MLC2v? From what is shown in Fig. 3, it seems that HOs mostly express either MLC2a or MLC2v. What are the numbers here? Fig. 3a shows strong expression of MLC2a but very little MLC2v. Fig. 3b shows a patch of MLC2a expressing cells and another patch that seems to express both MLC2a and MLC2v. Proper chamber specification as shown for a native mouse heart in the same figure looks quite different.

It is not very surprising that more complex HOs have a stronger similarity to native mouse hearts than simple emboid bodies, which not yet have submitted to cardiac differentiation. A better an more comprehensive assessment is necessary to reveal similarities and differences between HOs and native hearts. As pointed out before, dissection of atria and ventricles might help in this regard.

We are grateful to the reviewers for their critical comments and valuable suggestions that have helped us to improve our paper. As indicated in the responses that follow, we have carefully taken these comments and suggestions into account in the revised version of the manuscript.

Reviewer #1 (Remarks to the Author):

The manuscript entitled "In vitro recapitulation of embryonic heart structure and function via FGF4 and extracellular matrix" by Lee et al. describes a methodology resulting in the generation of cardiac organoid like structures from mouse embryonic stem cells. The authors used a combination of ECM modulation using laminin-entactin in conjunction with activation of FGF-4 to form a self-organizing organoid. The authors demonstrated the presence of chamber-like formation, chamber-specific gene expression, ultrastructural maturation, and the development of conduction tissues. They also identified the development of innervation and vascularization within the organoid, and demonstrated basic electrophysiological features of the generated organoids.

The generation of organoids that recapitulate embryonic cardiac development using an in-vitro system may be of great significance and with several applications ranging from basic developmental questions to assessment for drug toxicity and teratogenicity screening. Nevertheless, while the current study may be of great interest there are several key issues that need further evaluation to strengthen the scientific findings:

1. The reproducibility of the system is not clearly assessed throughout the manuscript. If this method is indeed effective and reproducible for each finding, the authors should clearly state the number of heart organoids (HOs) demonstrating the presented phenotype. For example, what percentage of HOs exhibited chamber-specific gene expression profile? What percentage had markers of cardiomyocyte maturation and sub-specification? How many included TRPM-4 expression? Innervation? Were their variation in the sizes of the different chamber-like structures?

Thank you for this valuable comment. According to reviewer's suggestions, we described the number of HOs with the presented phenotype in the manuscript. On the percentage of HOs exhibited chamber-specific gene expression, We added immunostaining data of Mlc2v and Mlc2a in HOs. 75% of HOs exhibited chamber specific expression such as segregated Mlc2a expression. We described it in the manuscript, line 339-341, "segregated expression of Mlc2a was detected in the most of cryosections of HOs (75%, n = 24, Fig. 5a, Fig. S5a), ..." and added data in Fig. 5a and Fig. S5a. Concerning cardiomyocyte maturation and sub-specification, 100% of iK1 and RyR expression was detected (n = 8). We described it in the manuscript, line 385, "iK1 expression in the HOs (100%, n = 8)", and line 265, "RYR2 were expressed in the cultured HOs (100%, n =8)". Also, 100% of TRPM4 expression was observed (n =14). We described it in the manuscript, line 249, "TRPM4 was expressed in all HOs (100%, n =14)". On the innervation, we added data of TEM analysis in HOs, at least 82% of HOs (n = 11) showed neuronal innervation. We described it in the manuscript, line 279-280, "...found in the HOs cultured for 11 days (at least 9 out of 11 HOs, Fig. S4b)", and new data were added in Fig. S4a. Regarding the variation in the sizes of the HOs, we added a sentence in the manuscript, line 136-138 "Although the sizes of HOs varied from 547µm to 1957µm, most HOs were over 700µm (average = 835.1µm, SEM = 29.9µm, n = 65, Fig. S2a, box plot)" and added box plot in Fig S2a.

2. The global HO gene expression analysis has nicely shown that HOs are more similar to the embryonic hearts than to EBs. Nevertheless, to further support the formation of different chamber types, regional (spatial) transcriptomics is required to ascertain chamber-specific gene expression patterns and reproducible and robust organoid self-organization. Thus the authors should use

laser-dissection or other methods to perform detailed gene-expression analysis of the different regions within the organoids (for example, from the regions identified in Fig.1E).

Thank you for this valuable comment. According to reviewer's suggestion, we performed RNA-seq analysis of two different regions (atrium-like and ventricle-like) within organoids by surgical dissection of organoids and embryonic hearts. We added new data in Fig. 5d and described it in the manuscript, line 363-369, "To further confirm the differential gene expression between the atrium- and ventricle-like regions in HOs, we performed RNA-seq analysis of the surgically dissected parts corresponding to these regions..."

3. In a similar manner, the investigators should use antibodies targeting multiple chamber-specific markers to further support the generation of distinct chambers.

Thank you for this valuable advice. According to reviewer's suggestion, we performed immunostaining analyses using antibodies targeting multiple chamber specific markers including Cx43, Cx45, GATA4, and cTnI. We added new data in Fig. 3b and described it in the manuscript, line 232-240, "Further characterization of the HOs with series of cardiac markers GATA4 (required for heart tube formation)³⁶, cTnI, Connexin43 (Cx43, GJA1)³⁷, Cx45 (GJCI)³⁷, cTnI and Nkx2-5 indicated that the HOs have similar cardiac features to embryonic hearts (Fig. 3b)..."

4. To ascertain that the presumed HO morphologies are identifiable and reproducible, embryological perturbations are required. Silencing or knockout of genetic components or pharmacological inhibition of signaling pathways associated with chamber formation and/or cardiac looping (e.g. FGF-8) and/or primary and secondary field heart formation (e.g. ISL-1 knockout) should be assessed to demonstrate the ability of the in-vitro model to recapitulate known perturbations from the in-vivo environment.

Thank you for this valuable advice. According to reviewer's suggestion, we produced *Tbx5* knockout ES cell lines and *Isl1* knockout ES cell lines by the CRISPR/Cas9 system, and we performed heart organoid generation experiments with above KO cell lines. Expectedly, knockout of *Tbx5* and *Isl1* perturbed the generation of heart organoids (i.e., the retardation of cardiogenesis). We added new data in Fig. 2g and Fig. S3 and described it in the manuscript, line 205-215, "To ascertain that this system can be used to analyze *in vitro* cardiogenesis process, we performed the HO generations using *Tbx5* and *Isl1* (*Isl1*, a marker for proper looping during heart development³³) knockout (KO) ESCs generated by the CRISPR/Cas9 system (Fig. S3a). Expectedly, three *Tbx5* KO ESC lines exhibited delayed morphogenesis or failure of HO formation..."

5. Do the organoid display the presence of endocardial and epicardial cell-layers and if yes, do they express the necessary properties of such cell types?

Thank you for this important comment. Considering this issue, because PECAM (CD31) positive cell-layers were in heart organoids, it is possible to think that heart organoids may display the presence of endocardial cell-layers. However, we could not conclude the presence of epicardial cell-layers because it is not clear in both HOs and embryonic hearts. Regarding endocardial cells, we added the sentence in the manuscript, line 230-231, "suggesting the presence of endocardial cell-layers in the HOs."

6. The authors show that gene expression associated with endodermal structures (liver, intestine, pancreas) are expressed in the HOs (in higher levels than in in-vivo embryonic hearts). A purity assessment is required to ascertain that the presence of non-cardiac cells is indeed negligible, this should be done using both immunostaining, FACS for derivatives of non-mesodermal germ layers and single cell gene expression analysis.

Thank you for this valuable comment. Considering this issue, we performed the additional experiment of immunostaining with α 1 fetoprotein (AFP ; a representative marker of liver) to determine whether heart organoids contain endodermal structure. As result, heart organoids and embryonic hearts did not show AFP expression while they showed cTnT expression. Although *Sox17* is an intestine marker, *Sox17* is also expressed in endocardium of heart during cardiac development, and *Foxa2* is also expressed in EB. These results suggest the insufficient inhibition of endodermal stem state. On *Pdx1*, repression of *Pdx1* occurred in day13 HOs. We added new immunostaining data in Fig. 3b, and described it in the manuscript, line 241-242 and 303-308, “No expression of a liver marker, α 1 fetoprotein (AFP), also suggests the absence of functional endodermal parts in the HOs.”, and “Higher expressions of other tissue markers, such as *Foxa2/Sox17*, markers of the liver and intestine, and *Pdx1*, a marker of the pancreas in some of the HOs, may represent insufficient repression of developmental stage-specific markers during HO generation because *Foxa2* is usually expressed in the EBs...”.

7. The electrophysiological assessment does not assess the electrophysiological properties of the organoid and rather demonstrate characteristic cardiac tissue properties (partial isoproterenol response-chronotropic but not inotropic and APD prolongation). It is recommended that the EP evaluation will focus on the differential properties of atrial/ ventricular activation. Presence of components of the conduction system (SA node, AV node), conduction velocity in and between the formed chambers and presence of conduction blocks at the different stages. For example SA node cells can be identified as NKX negative/troponin positive cells,

Thank you for this important comment. According to reviewer’s suggestions, we performed additional experiments on EP evaluation by optical mapping analysis and identification SA node cells by immunostaining. As results, we could detect atrial- and ventricular-like activation similar to the observation of optical mapping of adult whole heart and we also observed Nkx2-5 negative and cTnI positive, SA node-like cells in the HOs. We added new data in Fig. 6e and Fig. 3b, and described it in the manuscript, line 392-397 and 239-240, respectively, “Also, optical mapping using voltage-sensitive dye identified two distinct regions representing atrial- and ventricular-like activation. Activation of atrial-like region preceded that of other area, and was characterized by relatively shorter action potential...”, and “Additionally, the population of cTnI positive and Nkx2-5 negative cells (white arrows, Fig. 3b) suggests that the presence of sinus node in the HOs like the embryonic hearts.”.

8. Connexin staining should be performed to evaluate the presence of gap junctions and connexin types (Cx43, Cx40, Cx45) since the type of connexin may correlate with the different cell/tissue types (atrial, ventricular, and purkinje cells).

Thank you for this valuable advice because cardiac connexins are important for impulse conduction. According to reviewer’s suggestion, we performed immunostaining analyses using antibodies targeting markers including Cx43 and Cx45, described it in the manuscript, line 233-238, “Connexin43 (Cx43, GJA1)³⁷, Cx45 (GJC1)³⁷, cTnI and Nkx2-5 indicated that the HOs have similar cardiac features to embryonic hearts (Fig. 3b). Cardiac connexins, Cx43 (in atrial and ventricular working myocardial cells), and Cx45 (localized in His-bundle and bundle branches) are gap junction proteins required for impulse propagation in the heart³⁷. Notably, the differential localization of two connexins in HOs indicates the presence of different cell types with gap junctions.”, and added new data in Fig. 3b.

9. While the findings of the study is interesting in generating a organoid model with primitive chamber-like structures the discussion should be toned down since the structures derived are still far from recapitulating the in vivo heart.

Thank you for this valuable advice. According to reviewer's suggestion, we rewrote discussion toned down and we changed "atrial and ventricular structure in HOs" to "atrium- and ventricle-like structure in HOs" through the manuscript. Additionally, we changed title to "*In vitro* generation of functional heart organoids via FGF4 and extra cellular matrix.", instead of "*In vitro* recapitulation of embryonic heart structure..."

10. Some of the immunostainings are of low resolution or are only provided in low-magnification and do not allow to identify cellular interactions. For example, it is not clear whether vessel like structure are formed.

Thank you for this valuable comment. In response to the reviewer's comment, we have performed some additional experiments and added the immunostaining images. Also, we provide high resolution figures in this revision. On the issue "whether vessel-like structure are formed", we do not claim the formation of vessel-like structure in HOs in this manuscript.

Reviewer #2 (Remarks to the Author):

In the study, Lee et al. used mouse embryonic stem cells to established an in vitro heart organoid (HO) system, which is claimed to represent a "faithful biomimetic model of the developing heart". Generation of the HOs mainly followed established procedures but differs from previously described approaches by the addition of FGF4 to the culture medium and the use of commercially available LN/ET (laminin/entactin) gel. The authors observed the generation of different cardiac cell types (including atrial and ventricular myocytes, Purkinje fibers, endothelial and smooth muscle cells as well as neurons) and the formation of cystic structures, which seem to promote separation of atrial and ventricular myocytes. The authors performed various morphological, molecular (RNA-seq) electrophysiological measurements to analyze properties of HOs and characterize the state of cells within HOs compared to corresponding developmental stages during mouse embryonic development.

Recapitulation of the development of composite organs in vitro by mere self-organization is a very ambitious endeavor and probably not realistic when it comes to complex morphogenetic processes requiring the input of different cues (including mechanical forces and blood flow) provided by neighboring tissues and signaling centers. Generation of cystic structures that vaguely remind of cardiac chambers and contain different cardiac cell types is simply not enough to justify the authors' bold claim to recapitulate "embryonic heart structure and function". The authors make numerous additional statements, which are vastly exaggerated and not really supported by their findings, although the authors made some interesting observations that might move the field forward. The only observation, which I indeed found remarkable, was the "spontaneous" separation of what seem to be atrial and ventricular cardiomyocytes in the HO cysts, which I have not seen published before.

Specific remarks

The authors have used FGF4 and commercial LN/ET gels to generate HO cysts but the use of these ingredients is poorly justified and badly controlled. I am not aware of a specific function of FGF4 during heart development. Why did the authors use FGF4 and not another FGF? Do other FGFs (e.g. FGF2, 6, 10) have the same properties as FGF4 for HO formation? In fact, several other groups have used FGFs to promote differentiation of ES cell to cardiomyocytes. For example, Chun et al. (PlosOne 2010) used FGF10 to facilitate cardiomyocyte differentiation and also the FGFR inhibitor PD173074 (same as used in this study) to prevent cardiomyocyte differentiation. Hence, it is hardly unexpected

that inhibition of FGFR signaling prevents cardiomyocyte formation. I found it also confusing that the authors sometimes refer to FGF4 as a factor that promotes proliferation of cardiomyocytes and sometimes as a factor that restricts proliferation and promotes differentiation.

Thank you for this valuable comment. Considering this issue, we performed new experiments with FGF2 (bFGF) and FGF10. While bFGF displayed over 50% of heart organoid (HO) generation efficiency, FGF10 showed only 33%(8/24) of HO generation efficiency. These results may be explained because bFGF and FGF4 have high affinity for FGFR1 (IIIc) but FGF10 does not have affinity for FGFR1 (IIIc). We added new data in Fig. 1d (FGF10) and in Fig. 1Se (bFGF), described in the manuscript, line 119-125, “To test whether other FGF signals (i.e., FGF10 which facilitate the differentiation of cardiomyocytes²⁶) enable to generate HO like FGF4, we applied FGF10 and FGF2 (bFGF) for HO generation. FGF10 showed only 33% (8/24) of HO generation efficiency (such as chamber formation, Fig. 1d)...”, and we cited Chan et al. (PLoS ONE 2010). Also, to exclude the uncertainty (confusion) of this issue regarding FGFR1, we rewrote it in the manuscript, line 127-128, “PD173074 (an FGFR1 inhibitor) prevented the cultured EBs from developing into HOs and resulted in excessive proliferation of monolayer cells”.

The authors show that LN/ET gels provide an advantage for HO formation compared to coated chamber slides but no other comparisons were made. What happens when alternative ECM matrixes are used? How pure are the commercial LN/ET gels? Are there other components included? A more systematic evaluation of different matrix components would be very helpful to precisely define conditions favoring organoid formation.

Thank you for this comment. We agree that this is a very important point. Therefore, we performed HO generation by using alternative ECM, Matrigel. Given that the efficiency of HO formation decreased dramatically, it is possible to think that LN/ET gels is one of necessary conditions favoring organoid formation. Therefore we added new data in Fig. 1c and described it in the manuscript, line 114-118, “EBs cultured with alternative ECM, Matrigel, also exhibited low efficiency of HO generation with multiple chambers (8%, n=12, Fig. 1c), suggesting that nonembedded LN/ET complex enables the decreased mechanical hindrance that would otherwise inhibit cellular movement and prevent adequate patterning and self-organization”. On the purity of the commercial LN/ET gels, these materials (with Laminin111 and entactin in an equimolar ratio) are purified (more than 90% of purity) from an Engelbreth-Holm-Swam tumor. Although we agree reviewer’s comment that a more systematic evaluation of different matrix components would be very helpful., it was technically difficult to apply some laminin types for HO formation because the concentration of commercial laminins are low. (i.e., low concentration of LN221 was not effective for HO formation.)

It requires a lot of imagination to recognize the different stages of heart development (cardiac crescent, primary heart tube, turning, four-chambered heart) in the cystic structures presented by the authors. Unfortunately, I am lacking such skills. In Figure 1e, the authors labeled different areas of a poly-cyst as right/left ventricles and left/right atria. Why that? Only because some cells lack Tbx5 expression? A much more comprehensive characterization of the different parts of the organoids is required to arrive at such conclusions.

Thank you for this valuable advice. According to reviewer’s suggestion, we performed immunostaining analyses using antibodies targeting multiple chamber specific markers including Cx43, Cx45, GATA4, and cTnI. We added new data in Fig. 3b and described it in the manuscript, line 232-240, “Further characterization of the HOs with series of cardiac markers GATA4 (required for heart tube formation)³⁶, cTnT, Connexin43 (Cx43, GJA1)³⁷, Cx45 (GJC1)³⁷, cTnI and Nkx2-5 indicated that the HOs have similar cardiac features to embryonic hearts (Fig.

3b)...”. Also, for Fig. 1e, we changed labels to LA-like, LV-like, RA-like, and RV-like. Considering the issue “ more comprehensive characterization of the different parts”, we previously showed the data of different ultrastructural properties between atrium- and ventricle-like structures in HOs, and we also added new data of electrophysiological evaluation and gene expression profiling between the different parts of HOs in Fig. 6e and Fig. 5d.

In Figure 4b, the authors claim to see trabeculation of HOs. I can hardly detect anything in Fig 4b. Specific markers for trabeculation are available, which need to be used before making such claims.

Thank you for this valuable comment. To make it clear, we removed the claim for trabeculation of HOs.

The authors completely ignored the role of primary and secondary heart fields contributing to cardiac growth and morphogenesis, which was nicely addressed in a similar organoid study published in NC last year. Do the different heart fields properly contribute to cardiac development of HOs as in vivo?

Thank you for this important comment. We regret that we did not cite the report on precardiac organoids (Andersen et al, Nat. Commun. 2018). According to reviewer’s comment, we cited above research and described it in the manuscript, line 44-46, “including precardiac organoids with first heart field (FHF) and second heart field (SHF) contributing to cardiac development⁶”. On the contribution of two heart fields to cardiac development of HOs as in vivo, it is possible to think that FHF and SHF may contribute to cardiac morphogenesis in HOs because FHF-like (Tbx5 positive) and SHF-like (Nkx2-5) parts appeared in cardiac crescent-like structure of day 3 culture, and also *Tbx5* KO (defect of FHF) ESCs and *Isl1* KO (defect of SHF) ESCs could not form normal HOs showing chamber formation. Therefore, we described it in the manuscript, line 194-196, “Because FHF and SHF contribute to cardiac growth and morphogenesis⁶, it should be determined whether the HOs mimic heart development even from an early gestation stage in terms of specification of FHF and SHF. Then”, and line 201-202, “Tbx5 (FHF-like, Fig 2e) and Nkx2-5 (SHF-like, Fig. 2e) expression was observed in the appropriate regions of the EBs”, and added the label in Fig. 2e.

The authors claim that the current HO model is an excellent model for drug testing and for assessing the role of gene mutations for cardiac morphogenesis. Why didn’t the authors simply used available mutant mouse ESCs (e.g. *Isl1*, *Tbx5* or other) to substantiate the claim?

Thank you for this valuable advice. According to reviewer’s suggestion, we produced *Tbx5* knockout ES cell lines and *Isl1* knockout ES cell lines by the CRISPR/Cas9 system, and we performed heart organoid generation experiments with above KO cell lines. Expectedly, knockout of *Tbx5* and *Isl1* perturbed the generation of heart organoids (i.e., the retardation of cardiogenesis). We added new data in Fig. 2g and Fig. S3 and described it in the manuscript, line 205-215, “To ascertain that this system can be used to analyze *in vitro* cardiogenesis process, we performed the HO generations using *Tbx5* and *Isl1* (*Islet 1*, a marker for proper looping during heart development³³) knockout (KO) ESCs generated by the CRISPR/Cas9 system (Fig. S3a). Expectedly, three *Tbx5* KO ESC lines exhibited delayed morphogenesis or failure of HO formation...”.

The separation of atrial and ventricular cardiomyocytes in the HOs is interesting and the authors did a good job investigating ultrastructural features of atrial and ventricular cardiomyocytes. However, it is clearly required to subject the potential “atrium” and “ventricle” structures to a more comprehensive characterization by RNA-seq. If the structures are really as distinct as claimed by the authors, it should not be problem to dissect them and sequence the different “chambers” separately.

Thank you for this valuable comment. According to reviewer’s suggestion, we performed

RNA-seq analysis of different regions (atrium-like and ventricle-like) within organoids by surgical dissection of organoids and embryonic hearts. We added new data in Fig. 5d and described it in the manuscript, line 363-369, “To further confirm the differential gene expression between the atrium- and ventricle-like regions in HOs, we performed RNA-seq analysis of the surgically dissected parts corresponding to these regions...”.

In this context, I was also concerned that the authors wrote that weak and segregated expression of MLC2a was only visible in SOME HOs. In how many HOs did the authors observe strong and segregated expression of both MLC2a and MLC2v? From what is shown in Fig. 3, it seems that HOs mostly express either MLC2a or MLC2v. What are the numbers here? Fig. 3a shows strong expression of MLC2a but very little MLC2v. Fig. 3b shows a patch of MLC2a expressing cells and another patch that seems to express both MLC2a and MLC2v. Proper chamber specification as shown for a native mouse heart in the same figure looks quite different.

Thank you for this valuable comments. According to reviewer’s suggestions, we described the number of HOs with the segregated expression of both Mlc2a and Mlc2v in the manuscript. We added immunostaining data of Mlc2v and Mlc2a in HOs. 75% of HOs exhibited chamber specific expression such as segregated Mlc2a expression. We described it in the manuscript, line 339-341, “segregated expression of Mlc2a was detected in the most of cryosections of HOs (75%, n = 24, Fig. 5a top, Fig. S5a), ...” and added data in Fig. 5a and Fig. S5a. Considering the issue in Fig. 3b (Fig. 5b in this revised manuscript) “the patch that seems to express both Mlc2a and Mlc2v”, actually ventricles of embryonic hearts (E10.5 and E12.5 in Fig. 5a) showed both expression of Mlc2v and Mlc2a. Also we added data (another HO) in Fig. 5a.

It is not very surprising that more complex HOs have a stronger similarity to native mouse hearts than simple embroid bodies, which not yet have submitted to cardiac differentiation. A better an more comprehensive assessment is necessary to reveal similarities and differences between HOs and native hearts. As pointed out before, dissection of atria and ventricles might help in this regard.

Thank you for this valuable comment. We agree reviewer’s suggestion. We performed RNA-seq analysis of different regions (atrium-like and ventricle-like) within organoids by surgical dissection of organoids and embryonic hearts (Fig. 5d), and we also performed electrophysiological evaluation between the atrium-and ventricle-like regions of HOs (Fig. 6e), although it is technically difficult to do histological analysis using the dissected organoids because dissected parts are too small to make OCT blocks.

Reviewers' Comments:

Reviewer #1:

Remarks to the Author:

Lee et al. describe a methodology for the generation of self-organizing cardiac organoid like structures from mouse embryonic stem cells (ESC) using a combination of ECM modulation (LN/ET) and FGF-4. The authors demonstrated the presence of a chamber-like formation, chamber-specific gene expression, ultrastructural maturation, and the development of conduction tissues in the generated structures. The revised version of the manuscript also includes new immunostaining and RNA-seq data (characterizing the chamber-specific properties of the segregated atrial-like and ventricular regions), the use of TBX-5 and Isl-1 KO mouse ESC lines, evaluation of alternative ECM formula (Matrigel) and FGF isoforms, some quantification analysis, and functional data (optical mapping using voltage-sensitive dyes).

The revised version of the manuscript is significantly improved and the investigators attempted to address most of the reviewers' concerns. Nevertheless, some of the results of the new studies require further clarifications and some additional concerns remain.

(1) I think the most interesting observation described in the study is the spatial clustering of atrial and ventricular myocytes to specific regions. The new immunostaining and RNA-seq data of these regions (focusing on chamber-specific markers) and their comparisons to the embryonic heart are therefore important. However, not all the immunostaining data are convincing. The authors should provide, for example, high-resolution images for the MLC-2V and MLC-2A that allows to evaluate individual cells. Importantly, the connexins (Cx43/Cx45) immunostainings (presented in Figure 3) seems non-specific at all. One would expect a punctuate membrane staining rather the diffuses and intense cytoplasmatic staining shown.

(2) In addition, for the connexin staining, Cx40 immunostaining is important to define atrial muscle (probably more important than the Cx45 staining).

(3) For the RNA-seq data, the atrial/ventricular segregation in term of chamber-specific gene expression is less robust as in the embryonic hearts (probably due to contaminating cells) and this should be acknowledged also in the manuscript. The authors should better clarify the explanation regarding what is shown in Figure 5d (it is clearer for supplementary Figure 5). The authors should mention in the text the names of the genes that display the largest atrial/ventricular expression differences and also discuss the expression of classical atria- and ventricular- specific genes.

(4) The electrophysiological data lack quantification. For example, regarding the calcium studies, the authors should quantify and statistically compare the effects of isoproterenol on a number of parameters (rate and kinetics of the rise and decay of the signal) vs. baseline.

(5) Similarly, regarding the optical action-potentials (APs), the authors claim that the atrial APs are shorter but there is no quantification or statistical comparison to prove that. Finally, it is not clear from the optical maps and movies presented whether there is clear atrial propagation in both "atria" before activation proceeds to activate both "ventricles". If this was the case in how many HO this propagation pattern was noted. Was there a delay in activation between the chambers (analogous to AV node conduction)? What was the conduction velocity within each "chamber"? Finally, there is no quantification and statistical data with regards to the E-4031 studies.

(6) The authors claim that immunofluorescence staining revealed IK1 expression in the HOs (100%, n = 8) similar to that of postnatal more mature than the early embryonic hearts, no examples of the embryonic heart are shown and no quantitative comparison was made.

Reviewer #2:

Remarks to the Author:

Lee et al. have submitted a revised version of their manuscript, which deals with the formation of heart organoids. The authors added several new data sets, which substantially improved the study. In particular, they now compared the effects of FGF4 to FGF2 and FGF10 and evaluated the suitability of other matrices for organoid formation. It was interesting to see that LN/ET was far superior compared to the conventional Matrigel. Another major improvement is the use of Tbx5 and Isl1 knockout ES cell lines, which seem to generate defective cardiac organoids and the attempt to perform RNA-seq analysis, separately for atrial and ventricular structures.

Unfortunately, there are still some issues that bother me:

- 1.) The new RNAseq analysis of atrial and ventricular structures is not very telling and superficial. The authors only presented a bar plot (Fig. 5d) of a selected set of genes, which does not allow a clear statement how similar the atrial and ventricular structures developing in organoids are, compared to the natural counterparts. It is mandatory to provide a more systematic comparison and to indicate genes that in organoids follow the same pattern as in embryos as well as those that show a divergent pattern. Is there anything missing in organoids that is present in embryos and vice versa? To what degree recapitulates the transcriptome of atrial and ventricular structures the natural expression patterns in embryos? Such questions are easy to address by appropriate bioinformatical analysis.
- 2.) As mentioned above, I do appreciate the use of Tbx5 and Isl1 knockout ES cell lines, However, it is very difficult to detect the aberrant morphology of Tbx5 and Isl1 knockout organoids in Fig. 2D and Suppl. Fig. 3B. A simple immunostaining would have been much more informative than the depiction of mostly dark and amorphous structures.
- 3.) The quality of some images has somewhat improved. Yet, some figures are still problematic. E.g., I can hardly detect any TRPM4 staining in Fig. 3C, particularly in E10.5 embryos. Am I supposed to see TRPM4 signals in printouts? The staining only becomes visible, if I maximally enlarge the figure on my monitor. It should be easy enough to provide a suitable enlargement.
- 4.) The authors try to give numbers when describing the efficiency of the methods, but I found this very confusing. It is really cumbersome to scan through the manuscript for collecting the respective numbers. E.g., when describing the poor performance of FGF-2, FGF4 and FGF-10, the authors describe that FGF-2 treatment resulted in "over 50% heart organoid formation" compared to FGF-10 with 33%. What is the exact number for the FGF-2 treatment and where is the comparison to FGF-4? Is the effect of FGF-4 set as 100%? This is just an example. The authors need to find a way to display their findings in a precise manner, so that the reader can easily understand what is going on. It would be much better to provide a table comparing the efficiency of organoid formation in respect to the different parameters.
- 5.) I was slightly confused by the statement in the response letter that said: "we do not claim the formation of vessel-like structure in HO in this manuscript". On the other hand, the authors write about the formation of sinus venosus in line 324 and stain for arterial and venous markers. Is there vessel formation in the organoids or not?

We are grateful to the reviewers for their critical comments and valuable suggestions that have helped us to improve our paper. As indicated in the responses that follow, we have carefully taken these comments and suggestions into account in the revised version of the manuscript.

Reviewer #1 (Remarks to the Author):

Lee et al. describe a methodology for the generation of self-organizing cardiac organoid like structures from mouse embryonic stem cells (ESC) using a combination of ECM modulation (LN/ET) and FGF-4. The authors demonstrated the presence of a chamber-like formation, chamber-specific gene expression, ultrastructural maturation, and the development of conduction tissues in the generated structures. The revised version of the manuscript also includes new immunostaining and RNA-seq data (characterizing the chamber-specific properties of the segregated atrial-like and ventricular regions), the use of TBX-5 and Isl-1 KO mouse ESC lines, evaluation of alternative ECM formula (Matrigel) and FGF isoforms, some quantification analysis, and functional data (optical mapping using voltage-sensitive dyes).

The revised version of the manuscript is significantly improved and the investigators attempted to address most of the reviewers' concerns. Nevertheless, some of the results of the new studies require further clarifications and some additional concerns remain.

(1) I think the most interesting observation described in the study is the spatial clustering of atrial and ventricular myocytes to specific regions. The new immunostaining and RNA-seq data of these regions (focusing on chamber-specific markers) and their comparisons to the embryonic heart are therefore important. However, not all the immunostaining data are convincing. The authors should provide, for example, high-resolution images for the MLC-2V and MLC-2A that allows to evaluate individual cells. Importantly, the connexins (Cx43/Cx45) immunostainings (presented in Figure 3) seems non-specific at all. One would expect a punctuate membrane staining rather than the diffuses and intense cytoplasmic staining shown.

Thank you for this valuable advice. According to reviewer's suggestion, we performed additional immunostaining analyses using Mlc2v/Mlc2a, and Cx43/Cx45. We added new data with high resolution images in Fig. 5a and Fig. 3b and described it in the manuscript, line 356-358, "High resolution images also displayed Mlc2a positive individual cells in the atrium-like part and Mlc2v positive individual cells in the ventricle-like part in the HO (Fig. 5a)."

(2) In addition, for the connexin staining, Cx40 immunostaining is important to define atrial muscle (probably more important than the Cx45 staining).

Thank you for this valuable comment. According to reviewer's suggestion, we performed additional immunostaining analyses using Cx40 and Cx43 and could detect Cx40 expression suggesting the existence of atrial muscle in HOs. We added new data in Fig. 3b and described it in the manuscript, line 250-253, "In addition, Cx40 is mainly expressed in atria of the heart. Notably, the differential localization of three connexins indicates the presence of different cell types with gap junctions and the expression pattern of Cx40 suggests the existence of atrial muscles with proper gap junction proteins in the HOs."

(3) For the RNA-seq data, the atrial/ventricular segregation in term of chamber-specific gene expression is less robust as in the embryonic hearts (probably due to contaminating cells) and this should be acknowledged also in the manuscript. The authors should better clarify the explanation regarding what is shown in Figure 5d (it is clearer for supplementary Figure 5). The authors should mention in the text the names of the genes that display the largest atrial/ventricular expression

differences and also discuss the expression of classical atria- and ventricular- specific genes.

Thank you for this valuable comment.

Considering global gene expression profiles regarding insufficient the atrial/ventricular segregation in HO compared to the embryonic hearts, we agree this insufficiency may be caused by contaminating cells. According to reviewer's suggestion, we acknowledged it in the manuscript, line 391-393, "However, the global gene expression profiles regarding the atrial/ventricular segregation in HO is less robust as in E11.5H probably due to contaminating cells."

Additionally, we changed Fig. 5d with previous Fig 5S, and added explanation in the manuscript, line 385-391, "several gene expression patterns of atrium-like and ventricle-like regions within the HO were similar to those observed in the embryonic hearts at day 11.5, for example, *Col6a2*, *Gstm6*, *Slc7a7* showed the higher expressions in atria *in vivo* and atrium-like parts in HO, and *Cend1*, *Gja1(Cx43)*, *Sema3c*, *Cspg4*, *Lng1*, and *Snai3* showed the higher expressions in ventricles *in vivo* and ventricle-like parts in HO, although *Bmp10* was not expressed in both atrium- and ventricle-like parts in HO (Fig. 5d, top and Fig. S6b)."

Regarding the expression of classical atrial- and ventricular- specific genes, we added data in Fig. 5d (bottom) and discussed it in the manuscript, line 393-404, "Additionally, we investigated the gene expression profile of classical ventricular- and atrial specific genes in atrium- and ventricle-like parts in HO based on the differentially expressed genes (951 genes with $-1 < \log_2\text{-fold change} > 1$, adjusted P value < 0.05) between atria and ventricles of E11.5H. The ventricular specific genes such as *Myl2 (Mlc2v)*, *Myl3 (Mlc1v)*, *Hand1*, and *Myh10* exhibited higher expression in ventricle-like part of HO similar to that in ventricles of E11.5H, while the atrial specific genes such as *Myl7 (Mlc2a)*, *Kcnj3*, *Tbx5*, and *Nr2f2* (also known as a venous transcription factor)⁴⁹ displayed lower expression in atrium-like part of HO different to that in atria of E11.5H (Fig. 5d, bottom). This incomplete atrial specification in gene expression profiles at the atrium-like part within HO may be due to the decreased expressions of atrial TFs (*Tbx5* and *Nr2f2*) which promote expressions of many atrial specific genes in addition to the insufficient surgical dissection of the HO."

(4) The electrophysiological data lack quantification. For example, regarding the calcium studies, the authors should quantify and statistically compare the effects of isoproterenol on a number of parameters (rate and kinetics of the rise and decay of the signal) vs. baseline.

Thank you for this valuable advice. According to reviewer's suggestion, we performed additional calcium study and the data was quantified and statistically compared the effects of isoproterenol on beating rate and time constant of calcium decay. This new data added in Fig.6b and described it in the manuscript, line 413-415, "In addition, administering isoproterenol, a β adrenergic agonist, led to the increase of beating rate and the decrease of time constant of Ca^{2+} decay (n=6, $p < 0.05$ by paired t-test, Fig. 6b)".

(5) Similarly, regarding the optical action-potentials (APs), the authors claim that the atrial APs are shorter but there is no quantification or statistical comparison to prove that. Finally, it is not clear from the optical maps and movies presented whether there is clear atrial propagation in both "atria" before activation proceeds to activate both "ventricles". If this was the case in how many HO this propagation pattern was noted. Was there a delay in activation between the chambers (analogous to AV node conduction)? What was the conduction velocity within each "chamber"? Finally, there is no quantification and statistical data with regards to the E-4031 studies.

Thank you for this valuable advice. According to reviewer's suggestion, we performed additional the optical action-potentials (APs) analyses and the data was quantified and statistically compared.

To make it clear the atrial and ventricular propagation, we added new data of activation map in Fig. 6d and movie in Supplementary Video 6. Also, this propagation was observed in 13 of 22 HOs. We described it in the manuscript, line 425-432, “Importantly, optical mapping using voltage-sensitive dye identified two distinct regions representing atrium- and ventricle-like activation such as atrium-like propagation before ventricle activation (Supplementary Video 6). These two components (atrium- and ventricle-like) of activation were detected in 13 of 22 HOs (59%). Activation of the atrium-like region preceded that of other area, and was characterized by relatively shorter action potential duration (APD). The ventricle-like region was excited after the atrium-like region with substantial conduction delay, and showed longer action potential duration (Fig. 6d, left, center). The atrium-like regions of different HOs exhibited significantly shorter APD90 than the ventricle-like region (Fig. 6d, right).”.

Considering the existence of a delay in activation between the chambers (analogous to AV node conduction), we could not detect any delay like AV node conduction in HOs by optical mapping. Regarding the conduction velocity within each chamber, the measurement of conduction velocity was difficult in HOs given that technical limits including very small size of atrium-like region and the shape of HOs.

For the E4031 studies, we performed additional experiments, added new data of APD in Fig. 6e, and described it in the manuscript, line 435-436, “Treatment with 100 nM of E4031, an IKr blocker, caused significant prolonged APD90 (Fig. 6e, left top; graph, left bottom; representative image).”

(6) The authors claim that immunofluorescence staining revealed IK1 expression in the HOs (100%, n = 8) similar to that of postnatal more mature than the early embryonic hearts, no examples of the embryonic heart are shown and no quantitative comparison was made.

Thank you for this valuable comment. According to reviewer’s suggestion, we added new immunostaining data of embryonic hearts including E9.5H, E10.5H, and E12.5H in Fig. 6c. E9.5H (n = 1) and E10.5H (n = 2) did not show the expression of IK1, while E12.5H (n = 3) exhibited the expression of IK1 similar to HOs. We described it in the manuscript, line 419-423, “Immunofluorescence staining revealed IK1 expression in the HOs (100%, n = 8) similar to that of E12.5H and postnatal day1 (P1) heart, indicating that the HOs were developmentally more mature than the early embryonic hearts (E9.5H and 10.5H without IK1 expression) in terms of IK1 (Fig. 6c).”.

--

Reviewer #2 (Remarks to the Author):

Lee et al. have submitted a revised version of their manuscript, which deals with the formation of heart organoids. The authored added several new data sets, which substantially improved the study. In particular, they now compared the effects of FGF4 to FGF2 and FGF10 and evaluated the suitability of other matrices for organoid formation. It was interesting to see that LN/ET was far superior compared to the conventional Matrigel. Another major improvement is the use of Tbx5 and Isl1 knockout ES cell lines, which seem to generate defective cardiac organoids and the attempt to perform RNA-seq analysis, separately for atrial and ventricular structures.

Unfortunately, there are still some issues that bother me:

1.) The new RNAseq analysis of atrial and ventricular structures is not very telling and superficial. The authors only presented a bar plot (Fig. 5d) of a selected set of genes, which does not allow a clear

statement how similar the atrial and ventricular structures developing in organoids are, compared to the natural counterparts. It is mandatory to provide a more systematic comparison and to indicate genes that in organoids follow the same pattern as in embryos as well as those that show a divergent pattern. Is there anything missing in organoids that is present in embryos and vice versa? To what degree recapitulates the transcriptome of atrial and ventricular structures the natural expression patterns in embryos? Such questions are easy to address by appropriate bioinformatical analysis.

Thank you for this valuable comment.

Although, we performed clustering analysis with RNA seq data of dissected HOs and E11.5 hearts, global gene expression profiles displayed insufficient the atrial/ventricular segregation in HOs compared to the embryonic hearts, we agree this insufficiency may be caused by contaminating cells. We described it in the manuscript, line 391-393, “However, the global gene expression profiles regarding the atrial/ventricular segregation in HOs is less robust as in E11.5H probably due to contaminating cells.”.

Additionally, we changed Fig. 5d with previous Fig 5S, and added explanation of the genes that in HOs follow the same pattern as in embryonic hearts and the gene missing in HOs, in the manuscript, line 385-391, “several gene expression patterns of atrium-like and ventricle-like regions within the HOs were similar to those observed in the embryonic hearts at day 11.5, for example, *Col6a2*, *Gstm6*, *Slc7a7* showed the higher expressions in atria *in vivo* and atrium-like parts in HOs, and *Cend1*, *Gja1(Cx43)*, *Sema3c*, *Cspg4*, *Lng1*, and *Snai3* showed the higher expressions in ventricles *in vivo* and ventricle-like parts in HOs, although *Bmp10* was not expressed in both atrium- and ventricle-like parts in HOs (Fig. 5d, top and Fig. S6b).”.

Probably given that the smaller size of HOs than embryonic hearts and/or contamination of each part of cells, The transcriptome of dissected atrium- and ventricle-like parts did not perfectly match with natural expression pattern of embryonic hearts. However, regarding the expression of classical atrial- and ventricular- specific genes, we added data in Fig. 5d (bottom) and discussed it in the manuscript, line 393-404, “Additionally, we investigated the gene expression profile of classical ventricular- and atrial specific genes in atrium- and ventricle-like parts in HOs based on the differentially expressed genes (951 genes with $-1 < \log_2\text{-fold change} > 1$, adjusted P value < 0.05) between atria and ventricles of E11.5H. The ventricular specific genes such as *Myl2 (Mlc2v)*, *Myl3 (Mlc1v)*, *Hand1*, and *Myh10* exhibited higher expression in ventricle-like part of HOs similar to that in ventricles of E11.5H, while the atrial specific genes such as *Myl7 (Mlc2a)*, *Kcnj3*, *Tbx5*, and *Nr2f2* (also known as a venous transcription factor)⁴⁹ displayed lower expression in atrium-like part of HOs different to that in atria of E11.5H (Fig. 5d, bottom). This incomplete atrial specification in gene expression profiles at the atrium-like part within HOs may be due to the decreased expressions of atrial TFs (*Tbx5* and *Nr2f2*) which promote expressions of many atrial specific genes in addition to the insufficient surgical dissection of the HOs.”.

2.) As mentioned above, I do appreciate the use of *Tbx5* and *Isl1* knockout ES cell lines, However, it is very difficult to detect the aberrant morphology of *Tbx5* and *Isl1* knockout organoids in Fig. 2D and Suppl. Fig. 3B. A simple immunostaining would have been much more informative than the depiction of mostly dark and amorphous structures.

Thank you for this valuable advice. We agree that this is a very important point. According to reviewer’s suggestion, we performed additional immunostaining with *Tbx5/Nkx2-5* (at day 3 HOs), *Tbx5/cTnT* (at day 6 HOs), and α SMA/cTnT (at day 13 HOs) antibody sets in *Tbx5* KO- and *Isl1* KO-derived HOs. We added new data in Fig. 2g and Fig. S4a and described it in the manuscript, line 216-226, “To further confirm whether cardiac TFs and other cardiac markers are normally expressed in the early morphogenetic stages of *Tbx5* KO- and *Isl1* KO-derived HOs, we examined *Tbx5* and *Nkx2-5* expressions at day 3 and *Tbx5* and cTnT expressions at day

6. Expectedly, in the *Tbx5* KO-derived HO, *Tbx5* expression was distinctively decreased and the typical location of *Tbx5* and *Nkx2-5* was disrupted although *Nkx2-5* was expressed at day 3 (Fig. 2g, top left). In the case of *Isl1* KO-derived HO, the expression of *Tbx5* and *Nkx2-5* was not clearly separated at day 3 (Fig. 2g, top right). Moreover, both KO-derived HO showed delayed morphogenesis such as cardiac crescent like structure and also exhibited weak expression of cTnT³⁴ and disorganization of cardiac fibers in cTnT positive CMs compared to those of control HO at day 6 (Fig. 2g, bottom).”, and line 238-241 “Interestingly, the *Tbx5* KO-derived HO showed decreased expression of α SMA and cTnT while the *Isl1* KO-derived HO exhibited disorganized expression pattern of α SMA, i.e., outside the cTnT layer (Fig. S4a).”. Additionally, we transferred the photographs of *Tbx5* KO-derived HO to Fig. S3b, and added graphs and tables on the percentage of each morphogenetic step in *Tbx5* KO- and *Isl1* KO-derived HO in Fig. S3b.

3.) The quality of some images has somewhat improved. Yet, some figures are still problematic. E.g., I can hardly detect any TRPM4 staining in Fig. 3C, particularly in E10.5 embryos. Am I supposed to see TRPM4 signals in printouts? The staining only becomes visible, if I maximally enlarge the figure on my monitor. It should be easy enough to provide a suitable enlargement.

Thank you for this comment. According to reviewer’s suggestion, we changed the immunostaining image of TRPM4 in E10.5H with a suitable enlargement in Fig. 3c.

4.) The authors try to give numbers when describing the efficiency of the methods, but I found this very confusing. It is really cumbersome to scan through the manuscript for collecting the respective numbers. E.g., when describing the poor performance of FGF-2, FGF4 and FGF-10, the authors describe that FGF-2 treatment resulted in “over 50% heart organoid formation” compared to FGF-10 with 33%. What is the exact number for the FGF-2 treatment and where is the comparison to FGF-4? Is the effect of FGF-4 set as 100%? This is just an example. The authors need to find a way to display their findings in a precise manner, so that the reader can easily understand what is going on. It would be much better to provide a table comparing the efficiency of organoid formation in respect to the different parameters.

Thank you for this comment. According to reviewer’s suggestion, we added a graph on the chamber formation efficiency with FGF4 vs FGF10 in Fig. 1d, and described it in the manuscript, line 121-123, “FGF10 showed 42% (first experiment, 5/12), and 25% (second experiment, 3/12) of HO generation efficiency (such as chamber formation, Fig. 1d), while FGF4 displayed 88% and 78% of HO generation efficiency (Fig. 1d, right).”. Additionally, we added a table on the chamber formation efficiency with FGF4 vs bFGF (FGF2) in Fig. S1e, and described it in the manuscript, line 123-125, “In case of bFGF supplementation, HO generation efficiency was 58% (7/12) comparable to 78% (7/9) of FGF4 (Fig. S1e).”.

5.) I was slightly confused by the statement in the response letter that said: “we do not claim the formation of vessel-like structure in HO in this manuscript”. On the other hand, the authors write about the formation of sinus venosus in line 324 and stain for arterial and venous markers. Is there vessel formation in the organoids or not?

Thank you for this comment. Considering this issue, we detected only gene expression of arterial and venous marker genes in HO and did not performed immunostaining. Therefore, we deleted the comment for sinus venosus formation and changed sentences it in the manuscript, line 338-340, “Notably, the gene expression level of the venous marker *Nr2f2* in the HO was similar to that in *in vivo* embryonic hearts (Fig. S5c top). Similarly, in almost all of the HO (regardless of the culture period), the gene expression patterns of the artery markers..”.

Reviewers' Comments:

Reviewer #1:

Remarks to the Author:

The authors have addressed all my concerns and the revised manuscript is improved.

Reviewer #2:

Remarks to the Author:

Lee et al have further revised their manuscript, which describes an improved methodology to generate heart organoids from mouse embryonic stem cell-derived embryoid bodies. The authors have clarified a number of issues and added new data sets. The RNA-seq analysis has become more transparent and informative, although it turned out that there are substantial differences between the atrial and ventricular structures developing in the heart organoids and their natural counterparts. In my view it remains an open question whether this is due to contamination of other cells as written in the response letter or incomplete specification in organoid cultures. To give the authors credit: they now conclude that specification of the atrial part of the heart organoids is incomplete, which sounds much more realistic than the statements before. Importantly, the authors now also provide a much better depiction of the organoid morphology. Instead of whole mount images of mostly dark and amorphous structures, immunofluorescent staining of sections through the organoids were provided. The images demonstrate that the spatial expression of Nkx2.5 was disrupted in Tbx5 knock-out organoids, reflecting some aspects of Tbx5 mutant embryonic mouse hearts. Unfortunately, the Nkx2.5 staining is barely visible in the overview pictures of Tbx5 knock-out organoids but present in the magnification (Fig. 2g). The same applies for the Tbx5 staining in Isl1 KO organoids. Furthermore, the new Cx40 staining in Fig. 3b, which is supposed to demonstrate atrial specification, is very hard to detect. Same for IK1 in Fig. 6c. The authors should solve this issue, which can be easily done by increasing intensities of the respective fluorescence channels in overview pictures. Likewise, the staining intensities in Supplementary Figure S4b is pretty low, making it difficult to discern anything. In general, the presentation of the IF staining has somewhat improved but signals are still difficult to discern in some figures at regular magnification. This is somewhat annoying for a manuscript that underwent two rounds of evaluation, in particular since such issues can be easily fixed.

There is a typo in Supplementary Fig. 1e: It is "chamber" not "chameber". The font size is too small to read in some figures (e.g. Fig. 1d, Fig. 3d). Abbreviation for the inward rectifier potassium channel is IK1 and not iK1 as written in Fig. 6c. The authors should carefully edit the manuscript to eradicate such errors.

We are grateful to the the reviewer #2 for valuable comments and suggestions. As indicated in the responses that follow, we have carefully taken these comments and suggestions into account in the final version of the revised manuscript.

Reviewer #1 (Remarks to the Author):

The authors have addressed all my concerns and the revised manuscript is improved.
Thank you for this comment.

Reviewer #2 (Remarks to the Author):

Lee et al have further revised their manuscript, which describes an improved methodology to generate heart organoids from mouse embryonic stem cell-derived embryoid bodies. The authors have clarified a number of issues and added new data sets.

The RNA-seq analysis has become more transparent and informative, although it turned out that there are substantial differences between the atrial and ventricular structures developing in the heart organoids and their natural counterparts. In my view it remains an open question whether this is due to contamination of other cells as written in the response letter or incomplete specification in organoid cultures. To give the authors credit: they now conclude that specification of the atrial part of the heart organoids is incomplete, which sounds much more realistic than the statements before.

Thank you for this valuable comment.

We agree with your opinion and clearly stated the potential limitation of our method in the manuscript, “This discrepancy in gene expression profiles at the atrium-like part may represent incomplete atrial specification in organoid culture presumably due to the decreased expressions of atrial TFs (*Tbx5* and *Nr2f2*) which promote expressions of many atrial specific genes, or the insufficient surgical dissection of the heart organoids.”.

Importantly, the authors now also provide a much better depiction of the organoid morphology. Instead of whole mount images of mostly dark and amorphous structures, immunofluorescent staining of sections through the organoids were provided. The images demonstrate that the spatial expression of Nkx2.5 was disrupted in Tbx5 knock-out organoids, reflecting some aspects of Tbx5 mutant embryonic mouse hearts. Unfortunately, the Nkx2.5 staining in is barely visible in the overview pictures of Tbx5 knock-out organoids but present in the magnification (Fig. 2g). The same applies for the Tbx5 staining in Isl1 KO organoids. Furthermore, the new Cx40 staining in Fig. 3b, which is supposed to demonstrate atrial specification, is very hard to detect. Same for IK1 in Fig. 6c. The authors should solve this issue, which can be easily done by increasing intensities of the respective fluorescence channels in overview pictures. Likewise, the staining intensities in Supplementary Figure S4b is pretty low, making it difficult to discern anything. In general, the presentation of the IF staining has somewhat improved but signals are still difficult to discern in some figures at regular magnification. This is somewhat annoying for a manuscript that underwent two rounds of evaluation, in particular since such issues can be easily fixed.

Thank you for this important comment. According to reviewer’s comment and suggestions, we improved immunofluorescence contrast in images and enlarged figures of Nkx2.5 staining in Fig. 2g (now in Fig. 4c), Cx40 staining in Fig. 3b (now in Fig 5d), IK1 in Fig. 6c (now in Fig. 9c) and Supplementary Figure S4b. Overall, we improved figure presentation by adding display items of the enlarged figures (now 10 figures). We hope all these changes will satisfy your request.

There is a typo in Supplementary Fig. 1e: It is “chamber” not “chameber”.

We corrected it.

The font size is too small to read in some figures (e.g. Fig. 1d, Fig. 3d).

We enlarged font size in Fig. 1d and Fig. 3d (now Fig. 6b).

Abbreviation for the inward rectifier potassium channel is IK1 and not iK1 as written in Fig. 6c. The authors should carefully edit the manuscript to eradicate such errors.

Thank you for your indication.

We amended it accordingly.